# A local tumor microenvironment acquired super-enhancer induces an oncogenic driver in colorectal carcinoma

Royce W. Zhou[1,2,3], Jia Xu[1,12], Tiphaine C. Martin [1,12], Alexis L. Zachem[1], John He[1,2], Sait Ozturk[1], Deniz Demircioglu [1], Ankita Bansal[1], Andrew P. Trotta[1], Bruno Giotti[4], Berkley Gryder [5], Yao Shen[1], Xuewei Wu[1], Saul Carcamo[1,2], Kaitlyn Bosch[1,2], Benjamin Hopkins [1,2,4], Alexander Tsankov [2,4], Randolph Steinhagen[6], Drew R. Jones[7], John Asara[8], Jerry E. Chipuk[1,2], Rachel Brody[9], Steven Itzkowitz[10], Iok In Christine Chio [11], Dan Hasson[1,2], Emily Bernstein [1,2] & Ramon E. Parsons [1,2] ✉

Tumors exhibit enhancer reprogramming compared to normal tissue. The etiology is largely attributed to cell-intrinsic genomic alterations. Here, using freshly resected primary CRC tumors and patient-matched adjacent normal colon, we find divergent epigenetic landscapes between CRC tumors and cell lines. Intriguingly, this phenomenon extends to highly recurrent aberrant super-enhancers gained in CRC over normal. We find one such super-enhancer activated in epithelial cancer cells due to surrounding inflammation in the tumor microenvironment. We restore this super-enhancer and its expressed gene, *PDZK1IP1*, following treatment with cytokines or xenotransplantation into nude mice, thus demonstrating cell-extrinsic etiology. We demonstrate mechanistically that *PDZK1IP1* enhances the reductive capacity CRC cancer cells via the pentose phosphate pathway. We show this activation enables efficient growth under oxidative conditions, challenging the previous notion that *PDZK1IP1* acts as a tumor suppressor in CRC. Collectively, these observations highlight the significance of epigenomic profiling on primary specimens.

Cells commit exceptional density of histone 3 lysine 27 acetylation (H3K27ac) and transcriptional machinery to large genomic regulatory regions termed super-enhancers (SEs)[1]. The criteria for SE calling are technical[1,2]. A regulatory element is defined as a super-enhancer using the Rank Ordering of Super-Enhancers (ROSE) algorithm, which calls SEs by stitching enhancer signal within 12.5 kb together (excluding promoters) into bins and rank ordering by signal density. While various enhancer marks may be used as input, including BRD4, MED1, or H3K4me1, H3K27ac remains the most common in human primary tissues given its economic signal-to-background ratio with limited starting material. Dense loci are postulated to regulate genes of particular importance, such as those for cell identity, transcriptional dependencies, or oncogenic drivers in the cancer setting[1,2]

First described in embryonic stem cells, SEs control expression of genes required for pluripotency, and dysregulation of the SE landscape occurs in cancer[1,2]. Identification of dysregulated SEs between primary tumors and patient-matched normal remains incomplete for many cancers, and downstream validation of tumor-enriched SEs for functional relevance for tumor growth is scarce.

The etiology of SE reprogramming in cancer is incompletely understood but is widely attributed to cancer cell-intrinsic genomic

**Fig. 1 | Recurrently dysregulated super-enhancers in CRC patients. a** Study overview. Figure adapted from SMART Servier Medical Art, reproduced with permission, licensed under a Creative Commons Attribution 3.0 unported license. **b** PCA of H3K27ac signal at 2026 SEs in CRC ($n = 15$ independent tissue samples), normal mucosa ($n = 15$), crypts ($n = 4$), and FAP adenomas ($n = 2$). **c, d** GSEA between SE proximal genes and differentially expressed genes between CRC and normal. **e** H3K27ac ChIP-seq track near *ASCL2*. Two proximal SEs are underlined. The y-axes of all ChIP-seq tracks are scaled the same. **f** 2026 SEs by log$_2$ fold change in H3K27ac signal with 12 candidate SE target genes based on overlap of ranking and recurrence annotated. **g** Heatmap of H3K27ac signal at 583 differentially expressed SEs ($P < 0.01$, two-sided Student's $t$ test). Source data are provided as a Source Data file. NES normalized enrichment score, FDR false discovery rate.

alterations. Non-coding SE can become focally amplified in multiple malignancies[3,4]. Somatic non-coding mutations that create a de novo *MYB* TF binding site resulting in formation of an oncogenic SE at *TAL1* has been described in T-ALL[5].

Whether the local tumor microenvironment influences the SE landscape of cancer cells is poorly understood. One previous study observed divergent enhancer landscapes between primary medulloblastoma tumors versus medulloblastoma cell lines, suggesting the possibility of local microenvironment influence[6]. However, the extent this phenomenon recapitulates enhancer reprogramming in cancer is unclear. Furthermore, it is unknown whether environment-induced enhancer changes are mere passenger events or functionally benefit tumor growth.

Here, we identify that one of the most highly recurrent SEs in CRC upregulates the expression of *PDZK1IP1* and that this is activated by inflammatory cytokines and the tumor microenvironment. First isolated based on its overexpression in epithelial cancers relative to normal tissue, *PDZK1IP1* (also known as DD96 and MAP17) encodes a 17 kDa transmembrane protein that stimulates the sodium-dependent uptake of mannose and glucose through the regulation of sodium-glucose linked transporter (SGLT) family and bypasses TNF-α dependent growth arrest[7–12].

In this work, we find *PDZK1IP1* overexpression or deletion in multiple model systems decreases growth of CRC xenografts more significantly than CRC cells in culture. We show *PDZK1IP1* overexpression increases glucose uptake and increases reductive capacity as measured by glutathione and NADPH. Our findings argue that the tumor microenvironment activates *PDZK1IP1* in colorectal epithelial cells where it acts as an oncogenic driver of tumor growth in collaboration with genetic drivers of the disease.

## Results

### Super-enhancer landscape of CRC and matched normal

CRC is the second leading cause of cancer-related deaths in the United States and one of the first cancers in which a divergent epigenomic landscape between tumor and normal was described[13–16]. We profiled SE landscapes in 15 CRC patients using H3K27ac chromatin immunoprecipitation and sequencing (ChIP-seq), with the aim of leveraging recurrently dysregulated SEs to identify dependency genes in CRC. To control for batch effect and patient heterogeneity, we processed each tumor in parallel with matched adjacent normal colon mucosa, totaling 30 unique tissue samples (Fig. 1a). We identified 2026 total SEs in tumor and normal, which accounted for approximately 50% of the H3K27ac signal, with 95% saturation of unique SE discovery at 10 and 11 patients for adjacent normal and primary CRC, respectively (Fig. S1a–c). H3K27ac deposition at these 2026 SEs exhibits tissue-of-origin specificity, clustering separately from other malignancies under hierarchical unsupervised analyses when re-processed together with our datasets using the same pipeline (Fig. S1d, see Methods section for accessions).

### Divergent super-enhancers between CRC and matched normal

To compare CRC and normal mucosa SE landscapes, we performed principal component analysis (PCA) of H3K27ac signal at all 2026 SEs. We observed distinct clustering of CRC and normal mucosa suggestive of divergent SE landscapes (Fig. 1b). As further validation, we included previous datasets from colon crypts harboring normal intestinal stem cells and Familial Adenomatous Polyposis adenomas[17]. Crypts clustered near normal mucosa and adenomas occupied the boundary between normal mucosa and CRC, corroborating a previously described enhancer transition state in these early lesions[17] (Fig. 1b).

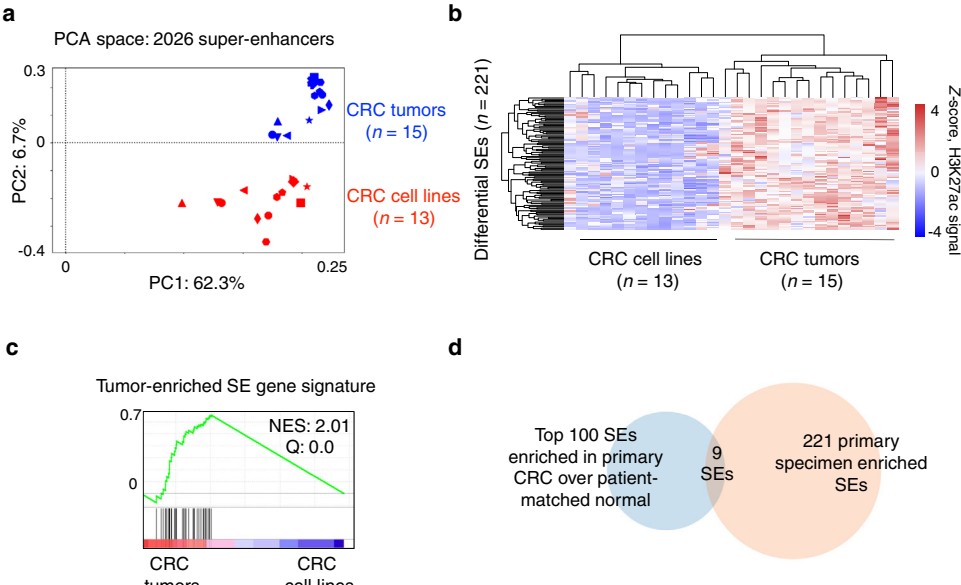

**Fig. 2 | A subset of super-enhancers is specific to primary CRC specimens and not recapitulated in CRC cell lines. a** PCA of H3K27ac signal at 2026 SEs in CRC ($n = 15$ independent tissue samples) and CRC cell lines ($n = 13$ independent cell lines). Please see Methods for accessions. **b** Heatmap of H3K27ac signal ($Z$-score) between CRC tumors and CRC cell lines with unsupervised hierarchical clustering and Pearson's correlation testing. 221 SEs (out of 2026) down-regulated in CRC cell lines compared to primary tumors (mean H3K27ac signal $\log_2$ fold change $<-1$,

$P < 0.01$). **c** GSEA validation of a tumor-enriched SE gene signature imposed on differentially expressed genes between primary CRC and CRC cell lines. See Supplementary Table 3 for gene list. **d** Venn diagram showing overlap between 221 SEs down-regulated in CRC cell lines from **b** with the top 100 gained SEs in primary CRC over patient-matched normal. NES normalized enrichment score, Q false discovery rate.

To identify dysregulated SEs between CRC and normal colon epithelium, we rank-ordered SEs by mean $\log_2$ fold-change in H3K27ac signal. SE changes corresponded with proximal gene expression enabling prediction of target genes (Fig. 1c, d and Supplementary Table 11). The vast majority of these predictions (84.2%) were supported by previously published Hi-ChIP data from primary CRC tumors (Supplementary Table 12)[16]. The most up-regulated SE in CRC (H3K27ac signal $\log_2$ fold change 2.73; $P = 0.0027$) was associated with *ASCL2*, a TF for intestinal stem cell fate (Figs. 1e and S2A)[18]. To prioritize SEs for downstream analyses, we overlapped 1) the top 100 SEs up-regulated in CRC with 2) SEs recurrent in 9 or more out of the total of 15 patients (Fig. 1f, g). We identified 12 candidate SEs enriched in primary CRC over adjacent normal with high recurrence in patients, including SEs assigned to genes previously implicated in CRC including Wnt target genes *MYC* and *ASCL2* (Figs. 1f and S2b).

**Unique super-enhancers in primary CRC over CRC cell lines**
Due to the technical challenges and cellular material required for high-resolution ChIP-seq, many continue to rely on commercially available cell lines as starting input for epigenomic studies. We investigated the extent commercially available CRC cell lines recapitulated the primary CRC tumor SE landscape, with the aim of identifying SEs that are unique to primary tumors.

To do so, we incorporated 13 commercially available CRC cell lines routinely used in laboratory studies, from previously published H3K27ac datasets (Supplementary Table 3)[17,19,20]. PCA analysis revealed distinct clustering of H3K27ac signal between primary CRC and CRC cell lines at all 2026 SEs along PC2 (6.7% of variation captured), including three CRC cell lines assayed in our laboratory as an internal control for potential batch effects (Fig. 2a). This observation was reproduced using independent H3K27ac ChIP-seq datasets derived from CRC tumors, CRC cell lines, and normal colon from a previously published study by Orouji et al. (Fig. S3a)[15]. This observation suggests some SEs in CRC may be dependent upon the physiological

environment within the living human body and not fully recapitulated in common laboratory cell line models of CRC.

In our dataset, we identified 221 SEs (~11%) as significantly underrepresented in CRC cell lines compared with primary CRC ($P < 0.01$ Student's $t$ test, H3K27ac signal $\log_2$ fold change $<-1$), hereafter referred to as "primary specimen enriched SEs" (Fig. 2b). 18 of these SEs were highly under-represented with a mean H3K27ac signal $\log_2$ fold change $<-2$. In contrast, only 12 SEs were significantly over-represented in CRC cell lines (Supplementary Table 4).

**Upstream TF regulation of primary specimen enriched SEs**
We next wished to better understand the TFs upstream of this SE discrepancy between primary CRC and CRC cell lines. We queried publicly available assay for transposase-accessible chromatin using sequencing (ATAC-seq) datasets from 81 patients with primary CRC from TCGA for motif analysis using the TRAP algorithm[21,22]. We found open chromatin at primary specimen enriched SEs to contain motifs of TFs with niche-dependent activity including NF-κB and STAT3, (Fig. S3b). Inflammatory NF-κB motifs were found at primary specimen enriched SEs but was not found in SEs found in both primary tumors and CRC cell lines (Fig. S3c). SEs boundaries are often flanked by CTCF sites for structural regulation, which validated in this analysis as the third most enriched motif.

We next assessed whether primary specimen enriched SEs had functional consequences on gene expression. We queried RNA-seq datasets from all 51 CRC cell lines from the Cancer Cell Line Encyclopedia (CCLE) in comparison with data from our primary CRCs[23]. Integrating our target gene predictions, we derived a gene signature downstream of primary specimen enriched SEs up-regulated in primary CRCs over CCLE cell lines (Fig. 2c, Supplementary Table 5).

Pathway analysis indicated an enrichment in genes involved in ion transport and homeostasis which are among the physiologic roles of the colon, as well as cytokine signaling (Fig. S3d)[24]. Furthermore, ARCHS4 algorithm analysis of these genes to predict upstream transcriptional regulation identify large intestine lineage factors *CDX1* and

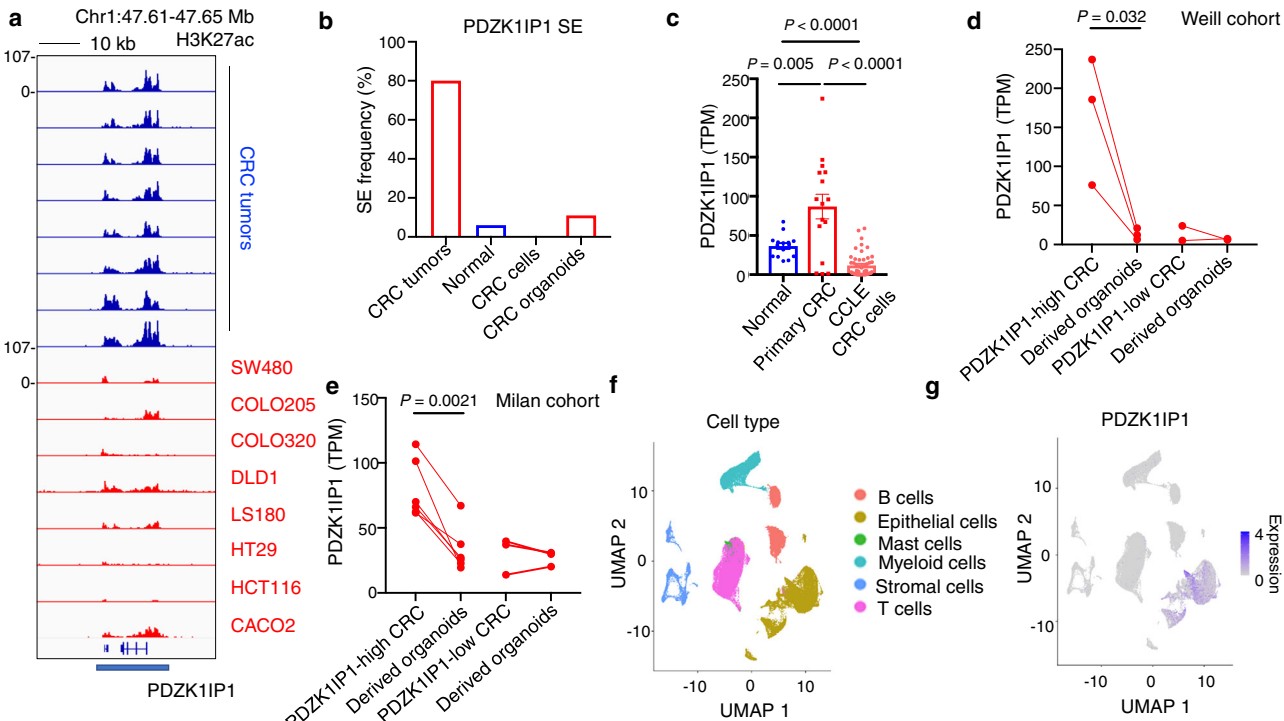

**Fig. 3 | A context-dependent, primary CRC-enriched super-enhancer at the locus of *PDZK1IP1*. a** H3K27ac ChIP-seq tracks at the *PDZK1IP1* SE in primary CRCs (*n* = 8 representative independent tumors) and CRC cell lines (*n* = 8 representative lines). The y-axes of all ChIP-seq tracks are scaled to the same range [0-107]. **b** Frequency of samples meeting ROSE criteria for SE calling at the *PDZK1IP1* locus. *n* = 15 independent CRC tumors, *n* = 15 independent patient-matched normal colon mucosae, *n* = 13 independent CRC cell lines, *n* = 9 independent 3D CRC organoids. **c** RNA-seq expression of *PDZK1IP1* in primary CRC (*n* = 16 independent tumors), normal mucosa (*n* = 15 independent tissue samples), and CCLE CRC cell lines (*n* = 51 independent cell lines). Data presented as median ± interquartile range. Significance was determined using two-sided Student's *t* test. **d, e** RNA-seq expression of *PDZK1IP1* in primary CRCs and sample-matched patient-derived 3D CRC organoids. For the Weill cohort, *n* = 3 for *PDZK1IP1*-high tumors and organoids, *n* = 2 for *PDZK1IP1*-low tumors and organoids. For the Milan cohort, *n* = 6 for *PDZK1IP1*-high tumors and organoids, *n* = 3 for *PDZK1IP1*-low tumors and organoids. Significance determined by two-sided Student's *t* test. Please see Methods for accessions. **f, g** Uniform Manifold Approximation and Projection (UMAP) of single cells from 23 independent primary CRC tumors and 10 independent adjacent patient-matched normal colons merged into a single plot, annotated with cell type and *PDZK1IP1* expression. Please see Methods for accessions. Source data are provided as a Source Data file.

*CDX2* as top hits (Fig. S3e)[25–27]. ARSHS4 analysis also corroborated TRAP motif analysis identification of *HNF4A* as a significantly discovered motif in primary specimen enriched SEs (Fig. S3e).

**Tumor microenvironment acquired super-enhancer at *PDZK1IP1***

Of the primary specimen enriched SEs, 9 SEs were among the top 100 SEs we found to be enriched in primary CRC over patient-matched normal colon (Fig. 2d and Supplementary Table 6). Intriguingly, two were identified in Fig. 1 as among the top recurrently dysregulated SEs in CRC tumors. One SE, associated with the gene *PDZK1IP1*, ranked second out of 2026 SEs for recurrence in CRC, meeting ROSE criteria for SE calling in 12 out of 15 primary CRC tumors, 1 out of 15 patient-matched normal colon epithelia, and 0 out of 13 CRC cell lines (H3K27ac signal $\log_2$ fold change −2.07, *P* = 1.7e10$^{-6}$; Figs. 3a, b and S4a). Enrichment of H3K27ac signal at the *PDZK1IP1* super-enhancer in primary CRC over normal colon as well as CRC cell lines was independently validated using the Orouji et al datasets (Fig. S4b, c). The other SE, adjacent to and most significantly associated with the expression of interferon induced transmembrane protein *IFITM1*, met ROSE criteria for SE calling in 9 out of 15 primary CRC tumors, 0 out of 15 patient-matched normal colon epithelia, and 0 out of 13 CRC cell lines (H3K27ac signal $\log_2$ fold change −2.89, *P* = 1.8e10$^{-5}$; Fig. S5a).

The SE at *PDZK1IP1* spans the entire gene body of *PDZK1IP1* as well as the long non-coding RNA LINC00853 (Fig. S4a). RNA-seq analysis of all primary CRCs in our cohort found LINC00853 expression to be significantly below 1 TPM, our absolute minimum cutoff for expression, whereas the mean RNA-seq expression of *PDZK1IP1* was 88.8 TPM.

Furthermore, portions of this SE have been previously shown to regulate TAL1 in hematologic contexts[28]. However, TAL1 is similarly not expressed in CRC or normal colon by RNA-seq (<1 TPM) suggesting the SE is not likely to be functionally relevant for TAL1. This observation suggests the H3K27ac and open chromatin peak at LINC00853 may serve as a distal regulatory region for *PDZK1IP1*. Indeed, previously published ChIP-seq datasets from primary CRCs shows H3K4me1 deposition at this region with looping contact to the *PDZK1IP1* promoter (Fig. S4d–g)[15,29].

We assessed *PDZK1IP1* mRNA expression between CRC cell lines from the CCLE and primary CRCs. We discovered a 21.5-fold higher median *PDZK1IP1* mRNA levels in primary CRCs over CCLE cell lines, suggesting widespread inactivation of *PDZK1IP1* expression in culture (Fig. 3c). CCLE cell line expression was lower than normal mucosa, suggesting that up-regulation of *PDZK1IP1* in CRC is microenvironment acquired (Fig. 3c). As proof-of-principle, high-expressing CRC surgical specimens significantly reduced *PDZK1IP1* mRNA levels when transferred to culture as 3D organoids in two separate cohorts[30] (Dr. Benjamin Hopkins, personal communication) (Fig. 3d, e).

The tumor microenvironment contains cancer cells as well as immune, stromal, and other non-cancer cells. We wished to assess the expression distribution of *PDZK1IP1* and *IFITM1* across these cell populations. Analysis of published single-cell RNA-seq analyses identified epithelial cancer cells exclusively expressing *PDZK1IP1* within primary CRCs (Fig. 3f, g)[31]. In contrast, *IFITM1* exhibited significant expression in stromal and immune cells in addition to epithelial cells (Fig. S5b–d). This suggests the *PDZK1IP1* expression discrepancy

between primary specimens and tissue culture is unlikely due to the exclusion of other cell populations but rather is an epithelial cell re-programming phenomenon. For this reason, we chose to focus on *PDZK1IP1* for downstream functional analyses.

The *IFITM1* expression pattern led us to consider the level contamination of bulk tumor H3K27ac ChIP-seq datasets with immune and stromal cell types, an unavoidable confounder when starting with primary tumor specimens. To this end, we performed de-convolution of cell type populations all primary CRCs for which sample-matched bulk tumor RNA was available using the xCell algorithm[32]. Our analyses showed a dominant epithelial cell signature for most tumors (Fig. S5e). Interestingly, the two cases predicted to have a low epithelial signature score were among the three cases that did not meet ROSE criteria for *PDZK1IP1* SE calling (Fig. S5e, see patients 17787 and 19282).

Single-cell RNA-seq also revealed *PDZK1IP1* expression in all four consensus molecular subtypes (CMS) of CRC (Fig. S5f, g)[31,33,34]. Expression was mostly enriched in CMS1-3, and especially prevalent in CMS3, previously described as a predominantly metabolic signature (Fig. S5f, g)[34]. Most CRC cases are sporadic, initiated by mutations in the *APC* tumor suppressor gene, which largely comprise CMS2 and CMS3[35]. *PDZK1IP1* single-cell expression is less prevalent among CMS4, in agreement with the *PDZK1IP1* SE absence in tumors predicted as highly stromal by xCell (Fig. S5e). Taken together, these data further support *PDZK1IP1* expression within intra-tumoral epithelial cells as a re-programming event.

### Unique super-enhancers in primary CRC over 3D CRC organoids
H3K27ac ChIP-seq datasets from human CRC 3D organoids were re-analyzed through our pipeline for comparison to our primary CRCs[30]. We similarly observed distinct clustering away from primary CRCs as well as from 2D cell lines (Fig. S6a). We identified 180 (~8.9%) SEs differentially expressed between 3D organoids and primary tumors (*P* < 0.01 Student's *t* test, H3K27ac signal absolute $\log_2$ fold change >1; Fig. S6b). Our data suggests 3D organoids improve recapitulation of the SE landscape of primary CRC at the 221 loci under-represented in commercial 2D CRC cell lines but nevertheless diverge from primary CRC (Fig. S6c). In contrast to 2D commercial cell lines, most of these loci (127 SEs) were enriched in organoids over primary CRC, including at TGF-β signaling gene *BMP4* (Fig. S6b, d).

Importantly, 3D CRC organoid culture validated the decreased H3K27ac signal at the *PDZK1IP1* ($\log_2$ fold change −0.48, *P* = 0.05) and *IFITM1* ($\log_2$ fold change −1.87, *P* = 0.0002) SEs observed in 2D culture (Fig. S6e, f). The findings corroborate the observed decrease in *PDZK1IP1* mRNA expression between patient-matched primary CRCs and derived 3D CRC organoids (Fig. 3d, e).

### Xenotransplantation induces the *PDZK1IP1* super-enhancer
We first assessed genomic alterations as a possible etiology of the *PDZK1IP1* SE. The SE resides on chromosome 1p which rarely exhibits amplification in 195 CRC cases from TCGA[36]. Inspection of H3K27ac reads within the *PDZK1IP1* SE did not reveal any recurrent mutations in CRC over patient-matched normal. Moreover, *PDZK1IP1* SE status failed to correlate with individual somatic protein-coding mutations at other loci, microsatellite instability (MSI), or tumor stage (Supplementary Tables 7–9). One normal colon epithelium case in our cohort met ROSE criteria for *PDZK1IP1* SE calling, further suggesting tumor initiating mutations may be dispensable for induction.

As a proof-of-principle that the *PDZK1IP1* SE is under cell-extrinsic control by the tumor environment, we grew the CRC cell line HT29, which expresses low levels of *PDZK1IP1*, in culture or as subcutaneous xenografts in nude mice (Fig. 4a). Remarkably, we observed increased H3K27ac deposition ($\log_2$ fold change 1.24) at the *PDZK1IP1* locus that met SE criteria in three independent xenograft tumors, but not parental cells grown in vitro (Fig. 4b). Although the H3K27ac antibody

employed reacts to both human and mouse epitopes, sequenced ChIP-seq reads were mapped only to the human genome to specifically capture signal from human CRC cells within the xenograft. Broadly, HT29 xenotransplantation failed to restore SE expression at other loci under-represented in cell lines, including at *IFITM1* ($\log_2$ −0.34-fold).

We next assessed the effects of SE induction on *PDZK1IP1* expression, using antibodies and quantitative real-time PCR (qRT-PCR) primers specific to human *PDZK1IP1* to exclude any signal contribution from mouse cell xenograft tumor infiltrates. HT29 xenograft tumors exhibited a 34.4-fold increase of *PDZK1IP1* mRNA, which validated by immunoblot (Fig. 4c, d). To assess whether SE induction regulates other proximal genes, we performed RNA-seq on HT29 xenograft tumors and parental cells. Although previously published Hi-ChIP data shows the SE looping to genes in the vicinity, *PDZK1IP1* was the only differentially expressed protein coding gene within 500 kb of the SE (Figs. 4e and S4e–g). Importantly, HT29 xenograft tumors recapitulate *PDZK1IP1* mRNA and protein levels observed in most primary CRCs (Fig. S7).

### Super-enhancer CRISPR interference silences PDZK1IP1
We wished to target select regions of the *PDZK1IP1* SE using the repressor CRISPR dCas9-KRAB[37]. We used primary CRC ATAC-seq datasets and identified three open chromatin regions within the *PDZK1IP1* SE, one at the promoter (P), as well as two distal constituent enhancer sites termed E1 and E2 (Fig. 4f)[21]. Lentiviral dCas9-KRAB repression at any of these three sites silenced *PDZK1IP1* expression in HT29 cells in culture (Fig. 4g). When these lines were grown as xeno-graft tumors in nude mice, we found dCas9-KRAB repression to attenuate induction of PDZK1IP1 protein levels (Fig. 4g–j). As xenografts, the E2 region appeared more critical for expression compared to E1. Taken together, these findings definitively demonstrate *PDZK1IP1* as a target gene of this SE.

### The *PDZK1IP1* super-enhancer is regulated by inflammation
To better understand *PDZK1IP1* epigenetic regulation by the micro-environment, we queried bulk tumor ATAC-seq and RNA-seq datasets from TCGA[21]. We uncovered several enriched inflammatory TF motifs including NF-κB, STAT1, and STAT3, as well as CTCF at open chromatin in the *PDZK1IP1* promoter and SE (Fig. 5a–d).

We validated TF binding in various tissues and cell lines based on analysis of published ChIP-seq data (Fig. S8a, see Methods section for accessions). In support, gene set enrichment analysis (GSEA) demonstrated a strong correlation between *PDZK1IP1* mRNA expression and inflammatory signatures including TNFα, IFNγ, IL-6, and IFNβ signaling from TCGA (Fig. 5e–i). As proof-of-principle, stimulation of HT29 cells in vitro with the upstream pro-inflammatory cytokines TNFα, IFNγ, and IL-6, but not IFNβ, induced *PDZK1IP1* expression (Fig. 5j, k). Cytokines resulted in additive potentiation of *PDZK1IP1* expression suggesting these TFs act independently and cooperatively, and phenocopies levels observed in xenografts (Fig. 5j). Importantly, stimulation of HT29 in culture with TNFα, IFNγ, and IL-6 was also sufficient to induce the *PDZK1IP1* SE (Fig. 5l).

Multiple orthogonal approaches link inflammation in the setting of HT29 xenografts with *PDZK1IP1* induction. Consistent with our observations in human samples, we observed increased inflammatory TF signaling in HT29 xenografts relative to cells grown in culture. GSEA demonstrated enriched inflammatory response and NF-κB signatures in HT29 xenografts over parental cells (Fig. 5m). Next, the ChIP-X Enrichment Analysis (ChEA) algorithm identified NF-κB subunit RELA as the top TF central to up-regulated genes ($\log_2$ fold change >2) in HT29 xenografts over parental cells (Fig. S8b)[38]. In support of this data, gained enhancers in HT29 xenografts over parental cells ($\log_2$ fold change H3K27ac signal >1) similarly enriched for NF-κB and STAT3 motifs via the TRAP algorithm (Fig. S8c, d). Cytokine array profiling of

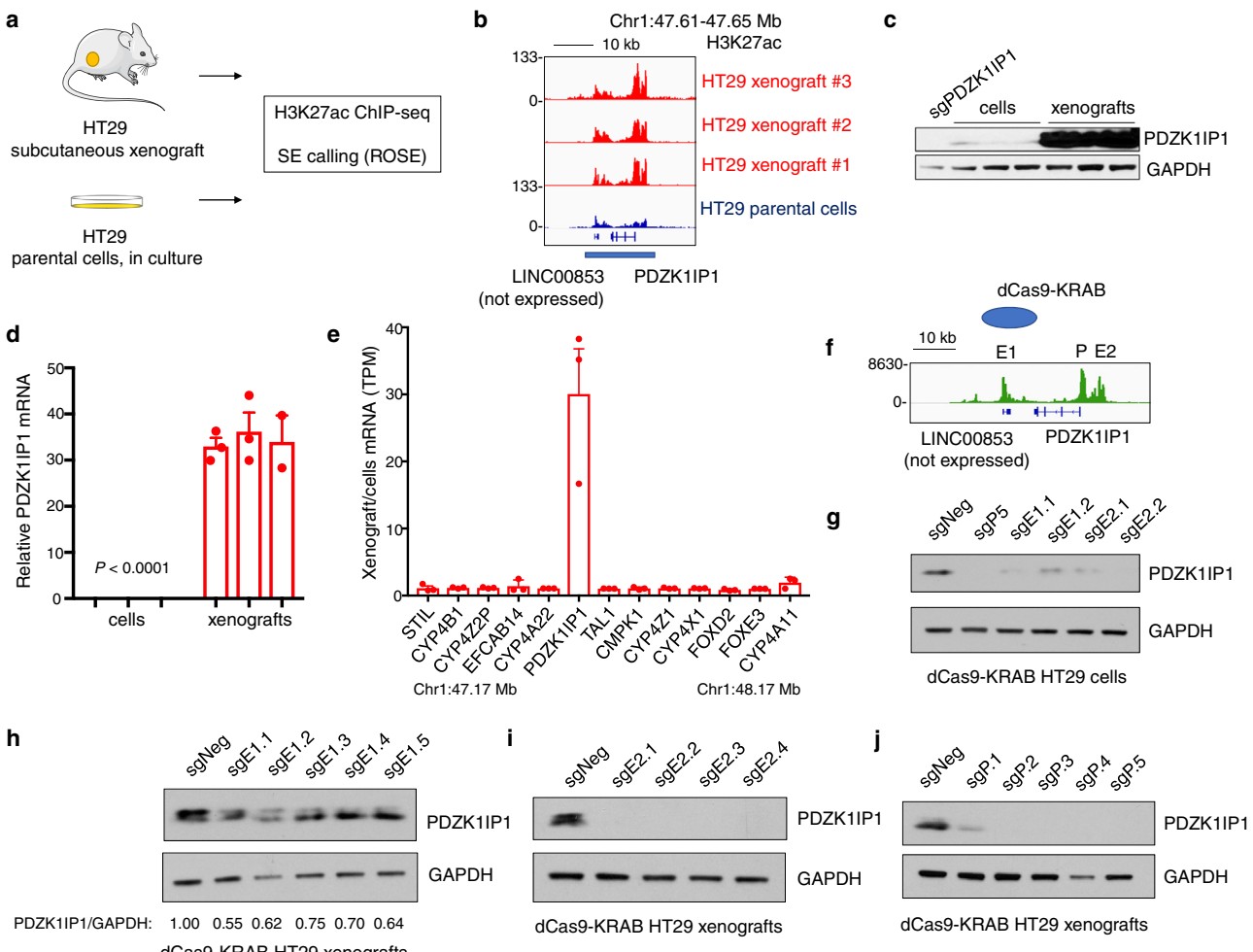

**Fig. 4 | Xenotransplantation restores the *PDZK1IP1* super-enhancer and CRISPR interference attenuates expression. a** Xenotransplantation and tumor ChIP-seq experiment overview. Figure adapted from SMART Servier Medical Art, reproduced with permission, licensed under a Creative Commons Attribution 3.0 unported license. **b** H3K27ac ChIP-seq track of the *PDZK1IP1* SE (underlined) in HT29 xenografts (*n* = 3 independent tumors) or HT29 parental cells maintained in culture. Y-axes of all ChIP-seq tracks are scaled to the same range [0–133]. **c** Immunoblot of *PDZK1IP1* protein levels between HT29 xenografts (*n* = 3 independent tumors) or HT29 parental cells maintained in culture (*n* = 3 biological replicates). **d** Quantitative real-time PCR (qRT-PCR) of *PDZK1IP1* mRNA expression between HT29 xenografts (*n* = 3 independent tumors) or HT29 parental cells maintained in culture (*n* = 3 biological replicates). Data presented as mean ± s.e.m. Significance was determined using two-sided Student's *t* test. **e** Fold change gene expression by RNA-seq between HT29 xenografts (*n* = 3 independent tumors) or HT29 parental cells maintained in culture (*n* = 3 biological replicates). X-axis represents all protein-coding genes within 500 kb on either side of the *PDZK1IP1* SE. Data presented as mean ± s.e.m. **f** dCas9-KRAB CRISPR interference of open chromatin regions at the PDZK1IP1 SE. ATAC-seq track at the *PDZK1IP1* SE in primary CRCs from TCGA (*n* = 81 independent tumors, merged into one track). **g–j** Immunoblot of PDZK1IP1 levels in HT29 cells expressing lentiviral sgRNAs with dCas9-KRAB targeting E1, E2, or P (promoter) regions of the *PDZK1IP1* SE, grown in culture or as xenograft tumors. Source data are provided as a Source Data file.

xenografts detected elevated human as well as murine pro-inflammatory cytokines and chemokines, including TNFα and IL-6 (Fig. S9a–c). Immunohistochemical analysis of HT29 xenograft tumors readily detected CD68+ and Ly6G+ cell staining, suggesting robust infiltration by macrophages and neutrophils or other granulocytes, respectively (Fig. S9d, e)[39,40]. Lastly, examination of previously published single-cell RNA-seq dataset from non-malignant inflamed colon from patients with inflammatory bowel disease show increased expression of PDZK1IP1 (Fig. S9f; see Methods section for accessions).

Importantly, deletion of NF-κB subunit RELA and STAT3 in HT29 cells inhibits PDZK1IP1 induction as xenografts (Fig. 5n, o). RELA and STAT3 are also required for PDZK1IP1 induction in culture in response to cytokines (Fig. S8e–i). In addition, we utilized CRISPR dCas9-KRAB to block STAT3 and NF-κB binding sites within the PDZK1IP1 SE, which similarly inhibited PDZK1IP1 expression (Fig. 5p–s). Taken together, these mechanistic experiments demonstrate RELA and STAT3 as required for cytokine and tumor microenvironment signaling to PDZK1IP1 induction.

CRC cells grown in culture exhibit decreased STAT3 signaling compared to primary tumors, and growing HT29 cells as a xenograft tumor reactivates this signaling[41]. Intriguingly, we and others validated this phenotype as unique to HT29 cells compared to COLO205 and DLD1, two other human CRC cell lines with low expression of *PDZK1IP1*[41]. COLO205 and DLD1 failed to phenocopy induction of *PDZK1IP1* protein levels, correlating with poor STAT3 reactivation in these lines (Fig. S10a). The finding that some cell lines lose the ability to respond to inflammation at the *PDZK1IP1* locus is perhaps explained by selection of known immune modulators to avoid growth inhibitory effects of inflammation[42].

In line with this finding, we did not observe increased H3K27ac signal in COLO205 liver metastasis xenografts in nude mice from orthotopic splenic vein injections from previously published data, nor in subcutaneous DLD1 xenografts in nude mice over respective parental lines maintained in culture performed for this study (Fig. S10a-c)[43]. Multiple other CRC cell lines we screened (including CACO2, SW48, SW480, SW620, RKO, and HCT116) exhibited no

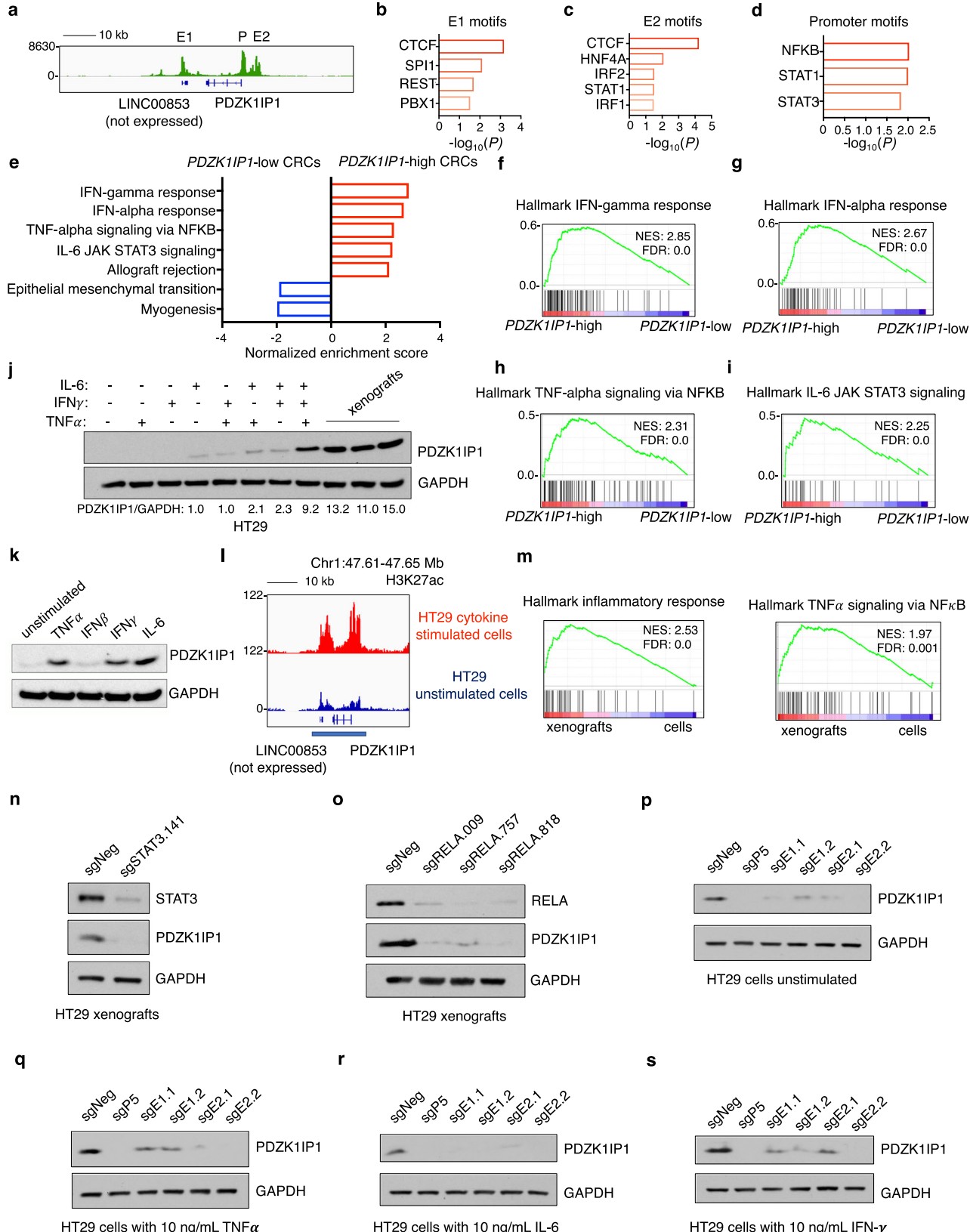

detectable expression of *PDZK1IP1* by RNA-seq or immunoblot. These cells failed to express *PDZK1IP1* in response to inflammatory cytokines or xenotransplantation (Fig. S10d). This suggests additional chromatin factors may be required to initiate *PDZK1IP1* expression to levels amenable to modulation by inflammation.

## Microenvironment recapitulates gained cancer enhancers

We next assessed 1) how much of enhancer re-programming in xenografts can be explained by inflammatory cytokines, and 2) how much of enhancer reprogramming in CRC can be explained by xenografting CRC cells to simulate a tumor microenvironment. A comparison between

**Fig. 5 | The *PDZK1IP1* super-enhancer is regulated by inflammation. a** ATAC-seq track at the *PDZK1IP1* SE in primary CRCs from TCGA (*n* = 81 independent tumors, merged into one track). **b**–**d** TRAP motif analysis at consensus open chromatin regions from the *PDZK1IP1* SE in primary CRC, where sequences are compared against all human promoters with a Benjamini-Hochberg correction to generate a *P*-value. **e** Hallmarks of Cancer GSEA of RNA-seq expression data from *PDZK1IP1*-high (top 50% mRNA expression) versus *PDZK1IP1*-low (bottom 50% mRNA expression) primary CRCs from TCGA (*n* = 342 independent tumors). **f**–**i** Hallmarks of Cancer GSEA enrichment signatures from *PDZK1IP1*-high (top 50% mRNA expression) versus *PDZK1IP1*-low (bottom 50% mRNA expression) primary CRCs from TCGA (*n* = 342 independent tumors). **j**, **k** *PDZK1IP1* expression levels by immunoblot in

cytokine stimulated HT29 cells (10 ng/mL, 16 hours) or HT29 subcutaneous xeno-grafts in nude mice. **l** H3K27ac ChIP-seq track of the *PDZK1IP1* SE (underlined) in unstimulated or TNFα, IFNγ, and IL-6 co-stimulated HT29 cells at 10 ng/mL for 16 hours. Y-axes of all ChIP-seq tracks are scaled to the same range [0-122]. **m** Hallmarks of Cancer GSEA performed on differentially expressed genes from RNA-seq between HT29 xenograft tumors (*n* = 3 independent tumors) and HT29 parental cells maintained in culture (*n* = 3 biological replicates). **n**, **o** Immunoblot of PDZK1IP1 levels in HT29 xenografts in the presence of WT or deleted RELA or STAT3. **p**–**s** Immunoblot of PDZK1IP1 levels in HT29 cells with the indicated treat-ment for 16 hours. Source data are provided as a Source Data file. NES normalized enrichment score, FDR false discovery rate.

HT29 xenografts and in vitro cytokine stimulated HT29 cells show a positive linear correlation of enhancer reprogramming, quantified as log$_2$ fold change in H3K27ac signal, at typical enhancers (*R* = 0.22), CTCF sites (*R* = 0.38), but not at promoters (*R* = 0.017) (Fig. S11a–c).

In addition, a comparison of HT29 xenografts and CRC showed modest correlation at typical enhancers (*R* = 0.11) but not at promoters (*R* = 0.03) (Fig. S11d-f). This trend validated in independent COLO205 peritoneal deposit and liver metastasis xenografts at typical enhancers (*R* = 0.19) (Fig. S11g–i)[43]. These findings more broadly implicate cyto-kines and the tumor environment as novel contributors to repro-gramming throughout the CRC enhancer landscape, predominantly at typical enhancers and CTCF sites.

### *PDZK1IP1* is an oncogenic driver in xenografts

*PDZK1IP1* (also known as MAP17, DD96) encodes a 114-residue protein overexpressed in multiple carcinomas[44–46]. High mRNA expression of *PDZK1IP1* negatively prognosticates patient survival in multiple can-cers including low-grade gliomas, glioblastomas, pancreatic adeno-carcinoma, but not CRC using data from TCGA (Fig. S12a–e).

Given the presence of a highly recurrent CRC-enriched and microenvironment acquired super-enhancer at *PDZK1IP1*, we assessed whether it reflects a transcriptional dependency for CRC growth. We deleted *PDZK1IP1* using two independent and non-overlapping CRISPR-cas9 guide RNAs (hereafter referred to as sgPDZK1IP1) and assessed the ability of human CRC lines to proliferate in vitro. We observed a very modest decrease in growth rate in HT29 cells whereas DLD1 cells remained unaffected using a 2D Incucyte proliferation assay over 6 days (Fig. S13a-d).

Given the observed induction of *PDZK1IP1* in vivo in HT29 cells, we grew the same lines as subcutaneous xenografts in nude mice. Remarkably, *PDZK1IP1* was critical for CRC tumor growth in vivo but not in vitro (Figs. 6a and S13e). The same trend is repeated in DLD1, which expressed *PDZK1IP1* at a much lower level, although the sgPDZK1IP1 growth defect was not as large as observed in HT29 (Figs. 6b and S13f).

We observed the same trend using gain of function models of *PDZK1IP1*. Given low levels of *PDZK1IP1* in vitro, we re-introduced V5-tagged *PDZK1IP1* (PDZK1IP1-V5) to protein levels observed in primary CRCs and HT29 xenograft tumors (Fig. S13g). PDZK1IP1-V5 and EV cells grew at similar rates in culture in HT29, COLO205, and DLD1 cells (Fig. S13h–j). When the same lines were grown as xenograft tumors, additional ectopic overexpression of PDZK1IP1-V5 sig-nificantly increased tumor growth in HT29 over empty vector con-trols (Figs. 6c and S13k). PDZK1IP1-V5 also significantly increased tumor growth of DLD1 (xenografts tumors for this model fail to endogenously induce *PDZK1IP1*) (Fig. 6d). Taken together, our data demonstrate *PDZK1IP1* is more important for tumor cell growth in mice, where it can be induced, than in tissue culture where expres-sion is low (Fig. 6e).

We next assessed whether CRISPR interference at the *PDZK1IP1* SE affects growth as xenograft tumors. We decided to target the E2 region as it most decreased PDZK1IP1 protein levels. Interference of the E2 region inhibited tumor growth compared to negative control, demonstrating that the SE itself regulates CRC growth (Fig. 6f).

### *PDZK1IP1* regulates cancer metabolism

The mechanism by which *PDZK1IP1* regulates tumor growth remains incompletely understood. Under normal physiology, *PDZK1IP1* is highly expressed in the proximal tubules of the kidney, where it is proposed to facilitate glucose reabsorption via the SGLT family of sodium-dependent glucose transporters[12]. Indeed, in a cohort of familial renal glucosuria primarily composed of patients with inacti-vating mutations in *SLC5A2* encoding the renal glucose transporter SGLT2, one patient was found to have a mutually exclusive homo-zygous truncating mutation in *PDZK1IP1*[12,47].

As previously observed, PDZK1IP1-V5 and EV cells grew at similar rates in culture and thus any metabolic phenotypes identified are independent of proliferation in this context (Fig. S13). However, *PDZK1IP1* over-expression has been reported to increase glucose uptake in breast cancer cells in a SGLT1 and SGLT2-dependent manner[48]. We observed increased intracellular glucose in PDZK1IP1-V5 overexpression cells compared to empty vector controls (Fig. S14a, c). Consistent with the literature, use of sotagliflozin, a SGLT1 and SGLT2 inhibitor, could rescue the increase in glucose uptake caused by PDZK1IP1-V5 (Fig. S14a, c)[49]. Sotagliflozin did not affect baseline glu-cose levels in EV cells, suggesting an inactivation of surplus glucose uptake. As an additional control, we performed a cellular uptake assay using 2-deoxyglucose (2DG), which is a poor substrate for SGLT family glucose transporters but is readily taken up by the GLUT family of passive glucose transporters. We observed no difference in 2DG uptake between PDZK1IP1-V5 and EV cells, further supporting the increase in intracellular glucose is due to SGLT glucose transporters (Fig. S14b, d).

The association of *PDZK1IP1* expression and metabolism gene signatures was independently noted in primary CRC tumors. *PDZK1IP1*-high primary CRCs from TCGA were enriched for glycolysis and oxi-dative phosphorylation gene sets by GSEA (Fig. S14e, f). In single-cell analyses, we previously observed CMS3 as the CRC subtype with the highest incidence of *PDZK1IP1* expression across 23 primary CRCs, which is defined by a highly metabolic gene signature (Fig. S5f, g)[34].

Thus, we considered whether *PDZK1IP1* affects downstream metabolism in CRC. A recent study demonstrated increased glycolytic rate in Seahorse live-cell assays with *PDZK1IP1* expression in HCC cells[50]. Surprisingly, in Seahorse live-cell assays, PDZK1IP1-V5 cells decreased extracellular acidification rate (ECAR), a read-out of glyco-lysis, and oxygen consumption rate (OCR), a read-out of mitochondrial respiration, compared to EV cells (Fig. S15a–d). We corroborated this result by performing a rapid glycolysis carbon tracing experiment using uniformly labeled C$^{13}$ glucose. We observed a significant decrease in M + 6 fructose 1,6 bisphosphate isotopologues in PDZK1IP1-V5 cells compared to EV (Fig. S15e, f)[51,52]. Decreases in ECAR and OCR did not significantly negatively impact cell proliferation or the ATP/ADP ratio (Fig. S15g, h).

We next performed global polar metabolite profiling by liquid chromatography-mass spectrometry to assess the effects of increased glucose uptake and decreased glycolytic rate on in CRC cells. We identified five consensus metabolites up-regulated in PDZK1IP1-V5 ( >1.25-fold, *P* < 0.05) in HT29, DLD1, and COLO205 lines. Intriguingly,

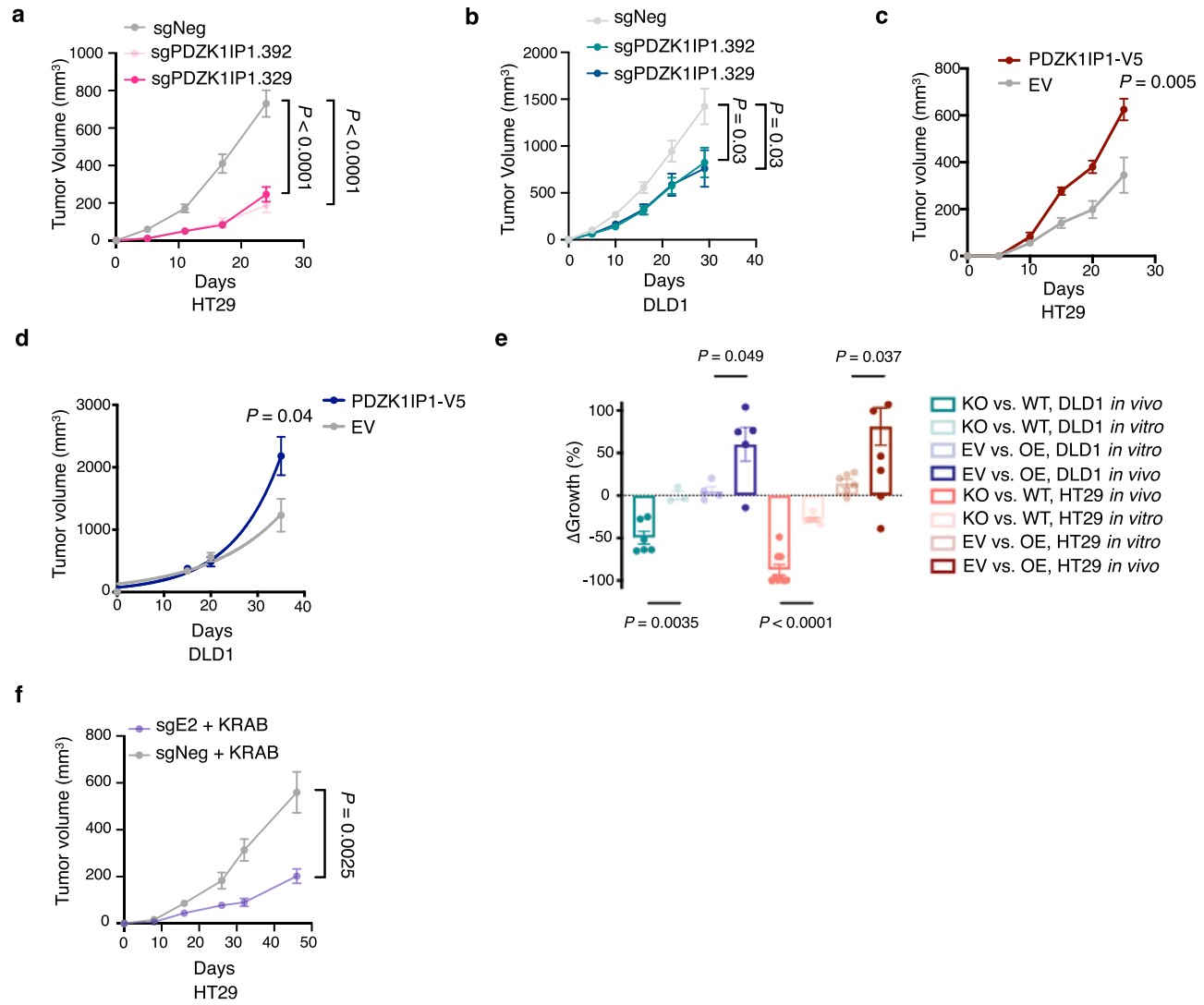

**Fig. 6 | *PDZK1IP1* is a context-dependent regulator of CRC tumor growth.**
**a, b** Growth curves, sgPDZK1IP1 and sgNeg subcutaneous xenograft tumors in nude mice. HT29 $n = 10$ mice per group, DLD1 $n = 10$ mice per group. 1 million cells injected per mouse for both HT29 and DLD1. Data presented as mean ± s.e.m. Significance was determined using two-sided Student's *t*-test. **c, d** Growth curves, PDZK1IP1-V5 and EV xenograft tumors. HT29 $n = 10$ mice per group, DLD1 $n = 5$ mice for PDZK1IP1-V5, $n = 8$ mice for EV. 3-5 million cells injected for both HT29 and DLD1. Data presented as mean ± s.e.m. Significance was determined using two-sided Student's *t* test. **e** Meta-comparison of 2D Incucyte growth curves in vitro with xenograft tumor growth in vivo. KO − sgPDZK1IP1.329, WT − sgNeg, EV − empty vector, OE − PDZK1IP1-V5. ΔGrowth presented as (KO-WT) as percentage of WT, or (OE-EV) as a percentage of EV. Data presented as mean ± s.e.m. Significance was determined using two-sided Student's *t*-test. HT29 KO and WT, as well DLD1 KO and WT in vivo ($n = 10$ mice per group). HT29 KO and WT, as well as DLD1 KO and WT in vitro ($n = 6$ biological replicates). **f** Growth curves of HT29 cells with CRISPR dCas9-KRAB interference of the E2 region of the *PDZK1IP1* SE (sgE2), and sgNeg subcutaneous xenograft tumors ($n = 8$ mice per group.) 1 million cells injected per mouse. Data presented as mean ± s.e.m. Significance was determined using two-sided Student's *t* test. Source data are provided as a Source Data file.

four of these—6-phosphogluconate, ribose 5-phosphate, IMP, and AMP—are intermediates or derivatives of the pentose phosphate pathway (PPP) (Figs. 7a, b and S16a). To assess increased PPP activity, we performed a carbon tracing experiment utilizing 1,2-C[13]-glucose. PDZK1IP1-V5 increased relative abundance of C[13]-labeled 6-phosphogluconolactone, the first and committed intermediate of the PPP in HT29, COLO205, and DLD1 lines (Fig. 7c–f).

The ratio of M + 1:M + 2 lactate is a readout of PPP flux relative to glycolysis[53]. We did not observe an increase in M + 1:M + 2 lactate between PDZK1IP1-V5 and EV cells (Fig. S16b–d). This suggests the increase in PPP activity is more likely due to increased glucose uptake with uniform distribution towards glycolysis and PPP, rather than shunting intermediates towards PPP at the expense of glycolysis.

The best understood role of the PPP in cancer is protection against reactive oxygen species (ROS)[54]. In this regard, we observed an increase in both NADPH/NADP + ratios and glutathione levels in PDZK1IP1-V5 cells at steady state (Figs. 7g-j and S16e). To assess whether this metabolic change in reductive capacity exhibits a cellular phenotype, we challenged cells with oxidative stress. We observed lower intracellular ROS levels in PDZK1IP1-V5 cells in response to the thiol oxidizing agent diamide in HT29 and DLD1 lines (Fig. 7k, l). Furthermore, PDZK1IP1-V5 exhibited increased cell viability when cultured under 1% O₂ or treated with diamide in HT29 lines (Fig. S16f–h). Taken together, we demonstrate *PDZK1IP1* protects against oxidative stress in CRC cells.

We next assessed whether *PDZK1IP1* regulation of PPP and reductive capacity was relevant towards the observed growth phenotype in vivo. Importantly, supplementing nude mice bearing sgPDZK1IP1 xenograft tumors with the antioxidant N-acetylcysteine (NAC) in drinking water fully rescued tumor growth (Fig. 7m). NAC did

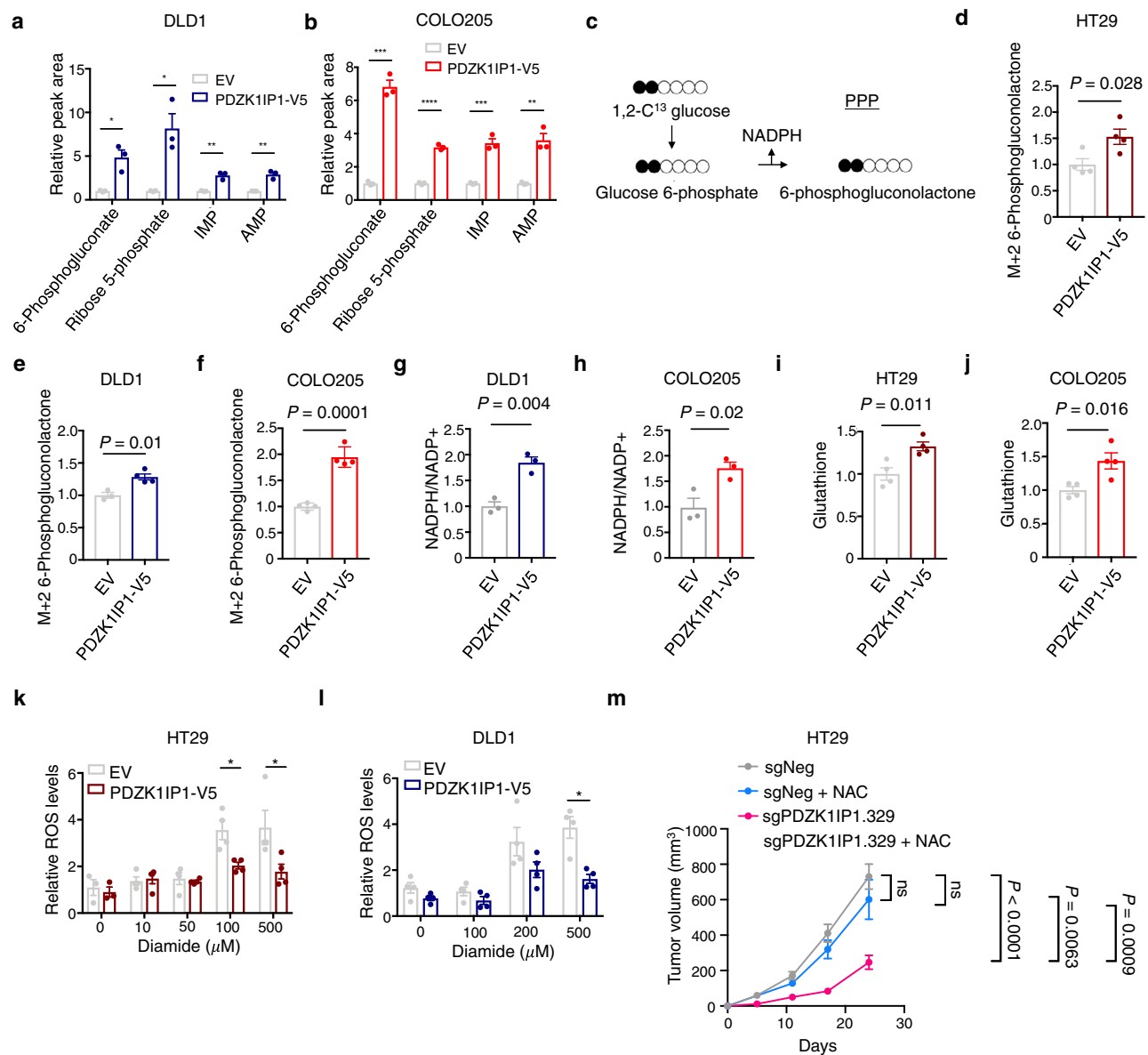

**Fig. 7 | *PDZK1IP1* regulates cancer metabolism. a, b** Steady state levels (total peak are) of PPP intermediates and downstream metabolites (*n* = 3 per group). Data presented as mean ± s.e.m. Significance was determined using two-sided Student's *t* test. **P* < 0.05, ***P* < 0.01, ****P* < 0.001, *****P* < 0.0001. **c** Schematic of PPP entry from glycolysis. **d–f** Total peak area of labeled 6-phosphogluconolactone (*n* = 4 per group), 1,2C¹³-glucose tracing. Data presented as mean ± s.e.m. Significance was determined using two-sided Student's *t* test. **g, h** Steady-state NADPH/NADP + ratios from total metabolite peak area (*n* = 3 per group). Data presented as mean ± s.e.m. Significance was determined using two-sided Student's *t* test. **i, j** Steady-state levels of total glutathione peak area (*n* = 4 per group). Data presented as mean ± s.e.m. Significance was determined using two-sided Student's *t* test. **k, l** Cellular ROS levels by luminescent assay following diamide treatment for 1 h at the indicated concentrations (*n* = 4 per group). Data presented as mean ± s.e.m. Significance was determined using Student's *t* test. * *P* < 0.05 **m** Growth curves of sgNeg and sgPDZK1IP1.329 xenograft tumors with or without 30 mM *N*-acetylcysteine in drinking water (*n* = 8 mice per group, 1 million cells injected). Data presented as mean ± s.e.m. Significance was determined using two-sided Student's *t* test. Source data are provided as a Source Data file.

not increase tumor growth of sgNeg cells, indicating the growth advantage is not general, but rather specific to the setting of *PDZK1IP1* loss (Fig. 7m). Taken together, our data suggest the role of PDZK1IP1 in regulating reductive capacity may be functionally relevant to facilitating CRC tumor growth.

## Discussion

Profiling the super-enhancer landscape has emerged as a powerful tool for uncovering novel target genes in cancer, particularly in those with few recurrently mutated genes[55]. The mechanisms by which super-enhancer reprogramming occur in cancer are not fully understood. The enhancer landscape of cancer cell lines has been observed to diverge from their cancer counterparts, suggesting the possibility of cell-extrinsic microenvironment influences on the epigenetic landscape that could contribute to tumor cell development and growth.

Here, we demonstrate the local tumor microenvironment induces super-enhancer reprogramming in CRC. We find the SE landscape of primary CRC to somewhat diverge (~11%) from in vitro CRC models, corroborating a previous observation in medulloblastoma[6]. Attempts to restore the tumor microenvironment in cell lines to model changes in the human epigenome are inherently flawed due to the requirement of immunocompromised mouse models. We expect the recapitulation of super-enhancer changes in this model to be limited given the notable absence of T cells in the microenvironment of our nude mouse model, especially at loci under regulation by inflammatory signaling

such as *PDZK1IP1*. This may explain failure of some cell lines to induce *PDZK1IP1* upon xenotransplantation. Nevertheless, its super-enhancer in HT29 cells is restored upon treatment with pro-inflammatory cytokines or xenotransplantation into nude mice, correlating with phosphorylation of STAT3.

We identify its target, *PDZK1IP1*, as a positive regulator of CRC growth. Oncogenic non-coding regions have been described but most occur in a small fraction of cases. We identified that the *PDZK1IP1* SE is among the most recurrently up-regulated SEs across 15 patient CRC samples, which is consistent with the common up-regulation of *PDZK1IP1* in epithelial carcinomas[45,46]. Activation of NF-κB and STAT3 signaling and transcriptional programs within cancer cells promote disease progression in multiple malignancies, including CRC[56–60]. Although regulation of *PDZK1IP1* expression by hypoxia was recently noted, we found its up-regulation to be negligible compared to levels induced by cytokines and xenotransplantation[50]. Our findings suggest that inflammation in the microenvironment is the primary mode of *PDZK1IP1* activation, likely as part of an adaptive response to signals that inhibit epithelial cell growth and viability.

In cancer cells, we find *PDZK1IP1* facilitates surplus glucose uptake via SGLT glucose transporters. SGLT2 inhibitors are FDA approved and undergoing Phase I/II clinical trials in solid tumors following promising pre-clinical results as monotherapy or in combination with PI3K or inhibitors or other therapies (canagliflozin: NCT04073680; dapagliflozin: NCT02695121, NCT04899349, and NCT04542291). Previous studies suggest *PDZK1IP1* increases ROS to contribute to cell proliferation. However, this does not agree with growing literature on *KRAS* and *BRAF* mutant tumors demonstrating rather it is the mitigation of ROS that benefits tumor growth in this context[61–64]. Moreover, *KRAS* and *BRAF* are among the most frequently mutated oncogenes in CRC[36]. We notably observed increased PPP activity, reductive capacity, and resistance to oxidative stress in HT29 (*BRAF* mutant), DLD1 (*KRAS* mutant), and COLO205 (*BRAF* mutant) cells. Our data show this function of *PDZK1IP1* endows tumor cells with a growth advantage in the tumor microenvironment but not in tissue culture. Thus, we elucidate a link between the context of *PDZK1IP1* super-enhancer activation and *PDZK1IP1* relevance to CRC growth.

It is unclear why our results conflict directly with previous other studies on *PDZK1IP1*, which report increased ROS levels as well as glycolytic and mitochondrial respiration rate[48,50]. While all studies over-expressed *PDZK1IP1*, they used different cancer cell lines with different lineage and mutation backgrounds to measure its effect. In addition, we chose a level of ectopic expression that mimics endogenous protein levels observed upon induction in xenografts as well as primary CRC tumors. Thus, a second possible explanation is that the other studies may be reporting the effect at different gene doses. Finally, ROS levels were measured by others using different probes.

Collectively, our observations broaden our notion of super-enhancers and highlights the importance of epigenomic profiling of primary tumor specimens and patient-matched controls. Our work further supports the need to assess and validate tumor-enriched super-enhancers to identify novel target genes. Antagonizing distal cell-extrinsic androgen or estrogen signaling to the cancer epigenome is a mainstay of initial therapy for hormonal cancers[65]. A greater understanding of local microenvironment factors and upstream intracellular signaling expands the scope for therapeutic strategies towards silencing cell-extrinsic enhancers and super-enhancers that control transcriptional dependencies in vivo. Targeted therapy that reduces inflammation to lower the level of expression of *PDZK1IP1* could be of benefit for patients with colorectal cancer.

## Methods

All research complies with relevant ethical regulations, including approval by the Mount Sinai Hospital Institutional Review Board (IRB) under Protocol IRB-19-01860 and the Mount Sinai Institutional Animal Care and Use Committee (IACUC) under Protocols #IACUC-2019-0048 and LA13-00024.

### Patient samples and power analysis

All human subjects work was approved by the Mount Sinai Hospital Institutional Review Board (IRB) under Protocol IRB-19-01860. Patient surgical specimens were resected by the Department of Colon and Rectal Surgery at Mount Sinai Hospital followed by pathologist separation into matched tumor and adjacent normal pairs. All samples were de-identified by The Mount Sinai Biorepository. For power analysis, total unique SEs for 15 primary CRCs and 15 adjacent normal mucosae were tallied. The percentage of total unique discovered SEs contributed by the acquisition of each patient sample, cumulative in the order in which the patient samples were received, was plotted.

### Cell lines

HT29, COLO205, DLD1, and HEK293T cells were purchased from the American Type Culture Collection (ATCC) and grown according to their specifications, supplemented with 10% fetal bovine serum (Thermo Fisher), 1% penicillin/streptomycin (Corning), and 5 μg/mL plasmocin prophylactic against mycoplasma (Invivogen). Furthermore, cells were tested for mycoplasma contamination at regular intervals to ensure mycoplasma-negative status (Lonza). All cell lines were maintained at 37 °C and 5% $CO_2$. Recombinant human TNFα, IFN-β, IFN-γ, and IL-6 were purchased from Peprotech. For cytokine stimulation experiments, cells were stimulated overnight for 16 hours.

### Animal experiments

In all, 6–8-week-old male nude mice (homozygous Nu/J) were purchased from The Jackson Laboratory (#002019). Male mice were used to better recapitulate CRC demographics. For HT29 xenograft experiments, 1 million cells suspended in 100 μL serum-free RPMI media were injected per nude mouse. During the HT29 N-acetylcysteine rescue experiment, 4 million cells were injected. For DLD1 xenograft experiments, 2 million cells suspended in 100uL serum-free RPMI media were injected per nude mouse. The maximum tumor volume permitted was 2000 $mm^3$. This was not exceeded in this study. For animal studies involving N-acetylcysteine (Sigma), drinking water supplemented with 30 mM NAC was used as the only drinking source. NAC solution was pH adjusted to 7.0 and refreshed weekly. Tumors were measured using calipers every 5 days, with volume calculated using the following formula: volume = 0.523(length)(width$^2$). All animal experiments were approved by the Mount Sinai Institutional Animal Care and Use Committee (IACUC) (protocol #IACUC-2019-0048 & LA13-00024). For mouse studies, no statistical method was used to predetermine sample size, mice were randomly distributed among the treatment groups, and no blinding was performed during data collection.

### Intracellular ROS and cell death

In all, –10,000 cells were plated in a 96-well plate overnight. Media containing 100 ng/mL doxycycline was refreshed the next day to induce expression of PDZK1IP1-V5 or empty vector (EV) for 48 h. Cells were treated with media containing diamide (Sigma) at the indicated concentrations for one hour. Intracellular ROS levels were determined using the ROS-Glo $H_2O_2$ assay (Promega) assay according to manufacturer's instructions before being read on a spectrophotometer. Viable cells after diamide treatment were assessed using the CellTiter 96 Non-Radioactive Cell Proliferation MTT Assay (Promega) according to manufacturer's instructions.

### Hypoxia chamber

For experiments involving hypoxia, 5000–10,000 cells were plated in a 96-well plate. Media containing 100 ng/mL doxycycline was refreshed the next day and the plate was moved into a HypOxystation H35

hypoxia chamber. The hypoxia chamber was maintained at 37°, humified, 5% $CO_2$, 1% $O_2$, and 94% $N_2$. Cells were kept in the hypoxia chamber for 96 h. Media containing 100 ng/mL doxycycline was maintained and refreshed every 48 h for EV or PDZK1IP1-V5 cells. Cellular assays were performed at this time point.

## Quantitative real-time PCR

RNA was isolated using the RNeasy Mini Kit (Qiagen). cDNA synthesis was performed using the iScript cDNA Synthesis Kit (Bio-Rad) according to manufacturer's instructions and analyzed by qPCR using SYBR green master mix (Life Technologies) on an ABI Prism 7500 instrument. 10 ng of cDNA was used per reaction. Relative expression was calculated using the $2^{-\triangle\triangle Ct}$ method as previously described[66]. For human PDZK1IP1, the following forward primer 5′- TCCTGACCGTCG-GAAACAAG −3′ and reverse primer 5′- TCGGGCACATTCTCATAGGC −3′ were used. For human GAPDH, the following forward primer 5′ - TCACCAGGGCTGCTTTTAAC − 3′ and reverse primer 5′ - AAT-GAAGGGGTCATTGATGG − 3′ were used.

## Immunoblot

Cell and tissue lysates were collected in RIPA buffer supplemented with with PhosSTOP and cOmplete protease inhibitors (Roche). Tissue samples were homogenized using a handheld TissueRuptor (QIAGEN). Samples were sonicated with a probe sonicator for 12 pulses before spinning down cellular debris at $12,300 \times g$ in a 4° centrifuge. Total protein concentrations were determined and normalized using a BCA assay (Thermo Fisher). Lysates were mixed 1:1 with 2x sample buffer (125 mM Tris-HCl at pH 6.8, 10% β-mercaptoethanol, 2% SDS, 20% glycerol, 0.05% Bromophenol Blue, 8 M urea) and boiled for 5 min. Protein lysates were resolved by electrophoresis using 4-12% Bis-Tris gels (Thermo Fisher). Samples were blotted on PVDF membrane (Millipore) by wet transfer. Membranes were blocked in 5% milk or 5% BSA in TBST for 1 h prior to overnight staining with primary antibody at 4 degrees Celsius with agitation. Membranes were rinsed three times with TBST followed by incubation with horseradish peroxidase-conjugated secondary antibodies in 5% milk for 1 h, and then rinsed again in TBST three times. Membranes were visualized using chemiluminescence (Thermo Fisher) and autoradiography film (Denville). The following primary antibodies were used for immunoblot: anti-PDZK1IP1 used 1:500 dilution (Sigma #HPA014907), anti-GAPDH used 1:10,000 dilution (Cell Signaling Technologies, CST #2118), anti-V5 used 1:1,000 dilution (CST #13202), anti-phosphorylated STAT3 at Y705 used 1:1,000 dilution (CST #9145), total STAT3 used 1:2,000 dilution (CST #9132), and NF-kB p65 encoded by RELA used 1:2,000 dilution (CST #8242). Band exposure was quantified using ImageJ. Horse radish peroxidase-conjugated secondary antibodies: mouse (Thermo #31432) and rabbit (Thermo #31460) both used at 1:2000 concentrations. Please see manufacturer's web page for internal and external validity on antibody validation.

## CRISPR-cas9 gene silencing

LentiCRISPRv2-puro (Plasmid #52961) and lentiCRISPRv2-blast (Plasmid #98293) were purchased from Addgene. Sense and antisense sgRNA oligos were synthesized using Fisher Scientific Eurofins, annealed, phosphorylated, and ligated into BsmBI (NEB) digested plasmids according to depositor's instructions as previously described[67]. The Quick Ligation Kit (NEB) was used to ligate DNA into the digested vectors. DNA was transformed into Stbl3 competent cells (Thermo Fisher) and ampicillin-resistant colonies were picked and screened for successful sgRNA integration by Sanger Sequencing (Genewiz) using the hU6-F primer (5′-GAGGGCCTATTTCCCATGATT-3′). For a list of sgRNAs used in this study, please see Supplementary Fig. 3.

## Ectopic overexpression of V5-tagged PDZK1IP1

The consensus coding sequence of *PDZK1IP1* was acquired from NCBI and PCR amplified from HT29 cell cDNA, derived as above. Inserts were cloned into the doxycycline-inducible pTRI-blas vector, a gift from Dr. Poulikos Poulikakos and Dr. Stuart Aaronson using the EcoR1 and BamH1 restriction enzyme sites (both from NEB). The *PDZK1IP1* carboxy-terminus comprises a PDZ binding domain. To minimize disruption of this domain which may require a free carboxy-terminus, the V5 peptide tag (GKPIPNPLLGLDST) was inserted in-frame immediately upstream of the PDZ binding domain. For a list of primers used in this study, see Supplementary Fig. 2. The Quick Ligation Kit (NEB) was used to ligate DNA into the digested vectors. DNA was transformed into Stbl3 competent cells (Thermo Fisher) and ampicillin-resistant colonies were picked and screened for successful sgRNA integration by Sanger Sequencing (Genewiz) using the pTRIPZ_seq_F primer (5′-GGCCATGGTGGCCCTCCTATAGTGA-3′). For doxycycline-induction of PDZK1IP1-V5 expression, cells were incubated with 100 ng/mL doxycycline for 48 hours prior to assay.

## Lentivirus production and cell transduction

Packaging plasmids psPAX2 (Plasmid # 12260) and pMD2.G (Plasmid # 12259) were purchased from Addgene. Lentiviral plasmids and packaging plasmids were assembled in Opti-MEM media (Corning) and transfected into HEK293T cells using the TransIT-Lenti Transfection Reagent (Mirus) according to the manufacturer's instructions. The mass ratio of psPAX2 to pMD2.G used for packaging was 3:1. Media was collected 48 hours after transfection and passed through a 0.45-µm filter and concentrated using Amicon Ultra-15 centrifugal filter (30 kDa, Millipore). Cells were transduced for 16 h followed by 48–72 h of puromycin selection or 6 days of blasticidin selection, depending on the vector. For puromycin, 2 µg/mL was used for HT29 cells and 3 µg/mL was used for DLD1 cells. For blasticidin, 5 µg/mL was used for HT29, COLO205, and DLD1 cells, refreshed every 72 h. We aimed for low MOI infections by empirically determining lentivirus dilution factors to achieve <50% cell survival during antibiotic selection.

## Chromatin immunoprecipitation followed by deep sequencing (ChIP-seq)

Xenograft tumors and patient surgical specimens were digested with 1X collagenase/hyaluronidase (Stem Cell Technologies), minced, and passed through a 100 µm filter. Cells were double crosslinked with 0.25 M disuccinimidyl glutarate followed by 1% formaldehyde. ChIP-seq was performed as previously described[68]. Briefly, 2X crosslinked chromatin was sonicated on a Diagenode Bioruptor for 15 cycles 30 s ON/30 s OFF on the low power setting. Cells were precleared using 100 µL of Magna-ChIP Protein A + G magnetic bead slurry (Millipore) before overnight incubation with 10 µg ChIP antibody conjugated to 100 µL beads. Beads were washed once with IP Wash Buffer I, twice with High Salt Wash Buffer, once with IP Wash Buffer II, and twice with TE buffer. For buffer recipes refer to Fontanals-Cirera, Hasson et al.[68]. Beads were eluted with 1% SDS at 65 degrees Celsius with agitation and reverse crosslinked overnight with high salt with RNase treatment (Roche). The next day, DNA underwent Proteinase K treatment at 42 degrees Celsius before elution using QIAquick PCR purification kit (QIAGEN). DNA was run on an Agilent High Sensitivity DNA chip using an Agilent Technologies 2100 Bioanalyzer for quality control. H3K27ac ChIP-seq antibodies were purchased from abcam (#ab177178).

## Incucyte in vitro cell proliferation assay

10,000 cells were plated in a 48-well plate (Corning Falcon). Cells were monitored for 5-6 days using an Incucyte live cell imaging system (Essen Bioscience), which is placed in a cell culture incubator operated at 37 degrees Celsius and 5% $CO_2$. For relevant experiments, 100 ng/mL doxycycline containing media was refreshed every 48 hours.

## DNA library construction and sequencing

DNA library construction was performed as previously described[68]. ChIP DNA was prepared as above. Briefly DNA underwent end repair, poly-adenylation, and barcoding using enzymes from New England Biolabs. ChIP libraries were amplified 12-15 PCR cycles using the 2X KAPA HiFi DNA HotStart ReadyMix before quality assessment by running on a High Sensitivity chip using a Bioanalyzer 2100 (Agilent) followed by a Qubit fluorometer (Thermo Fisher). The Mount Sinai Oncological Sciences Sequencing Core Facility prepared poly-A-capture RNA-seq libraries using the TruSeq RNA Library Prep Kit v2 (Illumina) according to manufacturer's instructions. All libraries were sequenced as 75 bp single end reads by the Mount Sinai Oncological Sciences Sequencing Core Facility on an Illumina NextSeq instrument.

## ChIP-seq data analysis and super-enhancer calling

Adapter sequences were removed from reads using Cutadapt. Reads were mapped to the hg19 human genome using bowtie. Duplicate reads were removed using samtools. Matching input control was used to call peaks. Peak calling was performed using MACS2. For H3K27ac, a $P$-value cutoff of $10^{-10}$ was used for peak calling. Bigwig tracks were generated using deepTools bamCoverage with RPKM normalization. H3K27ac ChIP-seq tracks were promoter normalized prior to direct comparison within and between patients. Blacklisted regions (Duke_Hg19SignalRepeatArtifactRegions.bed, downloaded from the Broad Institute) were excluded from called peaks using bedtools. Super-enhancer and enhancer calling were performed using Rank Ordering of Super-Enhancers (ROSE) on H3K27ac enrichment[69] using default parameters. Principal component analysis was performed using deepTools multiBigwigSummary and plotPCA functions. Summary plots were generated using deepTools computeMatrix and plotProfile. Pearson correlation heatmap and unsupervised hierarchical clustering analysis was performed using deepTools plotCorrelation. Student's $t$ test was used to call significant differentially expressed super-enhancers among the 2026 total identified between 15 primary CRCs and 15 patient-matched adjacent normal colon mucosa, using normalized H3K27ac RPKM as input. These significantly differential super-enhancers were visualized as a heatmap in Fig. 1 using the R package pheatmap on R version 4.0.3 and RStudio version 1.1.463. Please see Supplementary Table 10 for quality control metrics for ChIP-seq datasets generated in this study.

## RNA-seq data analysis

Sequenced reads were mapped to the hg38 transcriptome using Salmon quasi-mapping under the default parameters and expression output in TPM was extracted[70]. Output files (expression in TPM) were directly integrated into the R package DESeq2 for differential gene expression analysis, again under default parameters[71]. Genes with expression >2 TPM were used for downstream analyses. Pathway analyses were performed using pre-ranked Gene Set Enrichment Analysis (GSEA), using the pre-ranked (by log$_2$ fold change gene expression) function, as well as Enrichr (which includes many tools including ChEA)[72,73]. Log$_2$ fold change cut-offs for differentially expressed genes used in downstream analyses are stated in the text. For epithelial, immune, and stromal cell population de-convolution from bulk CRC tumor RNA-seq data, xCell was used with Salmon output and gene expression in TPM[32]. Out of the 15 CRC tumors, 11 had sample-matched RNA-seq data, for which xCell was performed to assess non-epithelial cell contrabution to H3K27ac signal from the bulk tumor ChIP-seq. The ratio of epithelial cell, immune cell, and stroma scores for each independent CRC is reported.

## Prediction of super-enhancer target genes

Sample-matched RNA-seq and H3K27ac ChIP-seq across multiple patient cohorts were utilized to predict SE target genes. For each of the 2026 SEs identified in this study, a linear regression analysis was performed, comparing natural variation of H3K27ac signal at the SE with sample-matched mRNA expression from RNA-seq for each protein coding gene with a transcriptional start site within 500 kb of the center of the SE. The gene with the most significant expression correlation ($P < 0.05$) is assigned as the most confident target gene. Several SEs have multiple target genes with significant correlation. Gene assignments with and without Hi-ChIP data accompaniment are listed in Supplementary Tables 1 and 2.

## ATAC-seq data and motif analysis

Previously generated ATAC-seq bigwig files mapped to human genome hg38 from primary colorectal carcinomas were downloaded, with permission, from TCGA[21]. LiftOver (UCSC) was used to convert coordinates between hg38 and hg19 human genomes. Tracks were visualized using the IGV browser[74]. Accessible chromatin regions of interest were subject to motif analysis using Transcription Factor Affinity Prediction (TRAP) web tools[22]. <500 bp regions were analyzed using Jaspar vertebrates as the matrix file, human promoters as the background model, and Benjamini-Hochberg as the multiple test correction. For TRAP motif analysis on multiple sequences, only the top 3000 sequences were analyzed per program input requirements.

## Single-cell RNA-seq analysis

Single-cell 3′ mRNA sequencing data from a cohort of 23 CRC patients (SMC cohort) was obtained from GSE132465[31]. R package Seurat (v.4.0) was used to analyze the data, using built in functions for normalization, dimensionality reduction, and visualization. To display cell clusters in UMAP plots and dotplots we colored and grouped cells by the annotations provided by the study, where malignant and non-malignant epithelial cells were combined into one category called epithelial cells. A customized version of the Seurat dot plot was used to display *PDZK1IP1* expression across cell types and patients.

## Mutation analysis of *PDZK1IP1* super-enhancer

H3K27ac ChIP-seq bam files for 15 patient tumor normal pairs were loaded into the IGV genome browser where the *PDZK1IP1* super-enhancer locus (chr1:47644434-47659486, hg19 coordinates) was visualized. Single nucleotide polymorphisms (SNPs), insertions, and deletions against the hg19 reference human genome within the super-enhancer locus were highlighted and assessed for specificity in CRC tumors over patient-matched normal in all 15 patients.

## Seahorse assays

Cellular glycolysis was measured using the XF Glycolysis stress test kit and XP Mitochondrial stress test kit (Agilent) according to the manufacturer's instructions. Briefly, PDZK1IP1-V5 and EV cells were seeded in XFe96 well plates (8000–10,000/well). The next day, media containing 100 ng/mL doxycycline was added to the cells for 48 hours to induce PDZK1IP1-V5 or EV expression. The next day, growth media was aspirated and replaced with Seahorse media (DMEM supplemented with 2 mM glutamine, pH 7.4) and cells were incubated in a were incubated at 37 °C in the absence of $CO_2$ for 1 h. Extracellular acidification rates (ECAR) and oxygen consumption rate (OCR) were measured using the XFe96 Extracellular Flux Analyzer. Baseline ECAR measurements were determined before administration of glucose (10 mM) and oligomycin (1 μM). Baseline OCR measurements were determined before administration of any drug. At the end of the assay, cells were stained with methylene blue, destained, and the absorbance was measured at 668 nm using a plate reader (Synergy H1 Hybrid multi-mode micro-plate reader, Biotek). ECAR and OCR measurements were normalized against the cell densities and all data analysis was conducted with the Seahorse Wave software version 2.6.1.53 (Agilent).

## Medical illustrations

Some figures as noted in the corresponding legends contain medical illustrations from SMART Servier Medical Art, reproduced with permission, licensed under a Creative Commons Attribution 3.0 unported license.

## Cytokine profiling

Cell line and tumor lysates were collected in RIPA buffer as described above, without the addition of 2X sample buffer. Supernatant lysates were diluted 1:1 with PBS supplemented with protease inhibitors. Cytokine profiling performed by Eve Technologies using the Human Cytokine Array/Chemokine Array 48-plex platform and Mouse Cytokine Array/Chemokine Array 31-plex panels. Cytokine concentrations were normalized to protein levels determined by BCA assay (Thermo Fisher).

## Targeted metabolomics and data analysis

For steady state polar metabolite profiling, full RPMI media was refreshed on cells for two hours. Polar metabolites were extracted from cancer cells in vitro using 80% (v/v) aqueous methanol as described before[75]. Targeted liquid chromatography-tandem mass spectrometry (LC-MS/MS) was performed using a 6500 or 5500 QTRAP triple quadrupole mass spectrometer (AB/SCIEX) coupled to a Prominence UFLC HPLC system (Shimadzu) with Amide HILIC chromatography (Waters). Data were acquired in selected reaction monitoring (SRM) mode using positive/negative ion polarity switching for steady-state polar profiling of greater than 260 molecules. Peak areas from the total ion current for each metabolite SRM transition were integrated using MultiQuant v2.0 software (AB/SCIEX). The original data were normalized to protein concentration determined by BCA assay.

## 1,2C[13] glucose carbon labeling

For carbon labeling experiments to assess the pentose phosphate pathway, cells were plated and PDZK1IP1-V5 induced in full RPMI. 24 hours prior to extraction, media was changed to RPMI with 5 mM glucose, 2 mM glutamine, and 10% dialyzed FBS as previously described[51]. Media was refreshed using RPMI with 5 mM glucose, 2 mM glutamine, and 10% dialyzed FBS for three hours to normalize metabolism. Media was then changed to RPMI with 5 mM 1,2C[13] glucose (Cambridge Isotope Laboratories), 2 mM glutamine, and 10% dialyzed FBS to allow carbon labeling for 4 h. Polar metabolites were extracted as above.

## Uniform C[13] glucose carbon labeling

For carbon labeling experiments to assess glycolysis, cells were plated and PDZK1IP1-V5 induced in full RPMI. 24 hours prior to extraction, media was changed to RPMI with 5 mM glucose, 2 mM glutamine, and 10% dialyzed FBS as previously described[51]. Media was refreshed using RPMI with 5 mM glucose, 2 mM glutamine, and 10% dialyzed FBS for three hours to normalize metabolism. Media was then changed to RPMI with 5 mM U-C[13] glucose (Cambridge Isotope Laboratories), 2 mM glutamine, and 10% dialyzed FBS to allow carbon labeling for 15 minutes. Polar metabolites were extracted as above.

## Immunohistochemistry

Xenograft tumors were fixed in 10% formalin (Thermo Fisher). IHC was performed by the Oncological Sciences Core Facility. Paraffin sections were dewaxed as previously described. Antigen retrieval was performed using a Dako pressure cooker in Antigen Unmasking Solution (Vector Laboratories). Briefly, sections were boiled in 0.01 M sodium citrate, pH 6.0, 0.05% Tween-20 twice for 10 min and cooled at room temperature for 20 min. Sections were washed with ddH$_2$O three times for 5 min, followed by a PBS wash for 5 min. HRP conjugated secondary antibody was used to detect signal. Endogenous HRP was blocked with 3% H$_2$O$_2$ in PBS or methanol for 10 min. The slide was washed in PBS three times for 5 minutes. The following primary antibodies were used for IHC: anti-CD11c (CST #97585), anti-CD68 (CST #97778), anti-perforin (CST #31647), and anti-Ly-6G (CST #87048).

## Glucose 6-phosphate dehydrogenase (G6PD) and 6-phosphogluconate (6PGD) activity assays

In vitro enzyme activity assays were performed as previously described[76]. Briefly, for glucose 6-phosphate dehydrogenase activity, the following substrates were re-constituted in vitro in PPP activity buffer (50 mM Tris, 1 mM MgCl$_2$): 200 μM glucose 6-phosphate (Sigma) and 100 μM NADP + (Roche). In all, 10 μg of protein lysate from PDZK1IP1-V5 or EV expressing cells were added to the reaction mixture. Reduction of NADP + to NADPH was read by absorption at 340 nm on a spectrophotometer as a readout of glucose 6-phosphate dehydrogenase activity. For 6-phosphogluconate dehydrogenase activity, the following substrates were re-constituted in vitro in. PPP activity buffer: 200 μM 6-phosphogluconate (Sigma) and 100 μM NADP + (Roche). 10 μg of protein lysate from PDZK1IP1-V5 or EV expressing cells were added to the reaction mixture. Reduction of NADP + to NADPH was read by absorption at 340 nm on a spectrophotometer as a readout of 6-phosphogluconate dehydrogenase activity.

## Statistics and data reproducibility

Statistical analyses were performed using Microsoft Excel for Mac Version 16 and Prism Graphpad Version 7. Data are expressed as mean with standard error of the mean unless otherwise noted. A Student's t-test (two-tailed with equal variance) was used when comparing two groups. A P value of <0.05 was considered statistically significant. No statistical method was used to predetermine sample size. We were not blinded to allocation during experiments and outcome assessment. Experiments in Figs. 4g–j and 5j, k, n–s were repeated independently three times with biological replicates; Supplementary Figs. 7b, c, 8e–i, 9d, e, 10c, d, 13c–g, and 13k have been repeated independently with similar results three times.

## Reporting summary

Further information on research design is available in the Nature Research Reporting Summary linked to this article.

## Data availability

H3K27ac ChIP-seq and RNA-seq datasets performed in this study are deposited at the NCBI Gene Expression Omnibus under the accession GSE166254. H3K27ac ChIP-seq datasets previously performed in FAP adenomas and normal colon crypts were accessed from the Gene Expression Omnibus under the accession GSE77737[17]. Broad Institute Cancer Cell Line Encyclopedia (CCLE) RNA-seq datasets for CRC cell lines were accessed from the Gene Expression Omnibus under the accession PRJNA523380[23]. COLO205 in vitro and orthotopic tumor H3K27ac ChIP-seq datasets were accessed under the accession GSE126188[43]. RELA ChIP-seq datasets from adipocytes were accessed under the accession GSM1566735[77]; RELA ChIP-seq datasets from A549 cells were accessed under the accession GSM847876[78]; STAT1 ChIP-seq datasets from HeLa cells were accessed under the accession GSM385505[79]; STAT1 ChIP-seq datasets from monocytes were accessed under the accession GSM1057011[80]; STAT3 ChIP-seq datasets from MDA-MB-468 cells were accessed under the accession GSM2278003[81]; CTCF ChIP-seq from normal colon tissue was accessed from ENCODE[82]. Osteosarcoma H3K27ac ChIP-seq datasets accessed under accession GSE74230[83]; gastric cancer H3K27ac ChIP-seq datasets accessed under accession GSE76153 & GSE75898[84]; luminal breast cancer H3K27ac ChIP-seq datasets accessed at the ENA under accession PRJEB22757[85]; prostate cancer H3K27ac ChIP-seq datasets accessed under accession GSE96652[86]; GBM H3K27ac ChIP-seq datasets accessed under

accession GSE119834[87]; chordoma H3K27ac ChIP-seq datasets accessed under accession GSE109794[88]; ccRCC H3K27ac ChIP-seq datasets accessed under accession GSE86095[89]; DLBCL H3K27ac ChIP-seq datasets accessed under accession GSE46663[90]. OncoLnc was used to generate Kaplan Meier curves for PDZK1IP1-high and -low mRNA expression cases based off TCGA data. TCGA public datasets used include ATAC-seq data and RNA-seq from all available CRC samples (listed as COAD for colon adenocarcinoma). Single-cell RNA-seq data from primary CRC and normal colon epithelium was accessed under the accession EGAS00001003779 and EGAS00001003769 from the European Genome-phenome Archive database[31]. Raw H3K27ac ChIP-seq sequencing data from primary CRC and matched human colon organoids, also referred to in the manuscript as the Milan cohort, were accessed under E-MTAB-8416, and raw RNA-seq sequencing data was accessed under E-MTAB-8448[30]. Raw H3K27ac ChIP-seq sequencing data from commercially available human CRC cell lines were accessed under GSE96069, GSE73319, GSE126188, and GSE77737[17,19,20,43]. H3K27ac and H3K4me1 ChIP-seq datasets previously published were accessed under the accession GSE88945)[15]. Hi-ChIP datasets previously published were accessed under the accession GSE133928; primary tumor samples MGH1904, MGH5328, MGH8416 from the original publication were used in this analysis[16]. The minimum dataset needed to interpret, verify, and extend the research as provided as Raw Source Data in the Supplementary Files; for larger datasets, please see NCBI Gene Expression Omnibus under the accession GSE166254. The following human genome accessions were used: The NCBI accession of the UCSC hg38 genome is GCA_000001405.15, and the UCSC hg19 genome is GCA_000001405.1. The following mouse genome accessions were used: GRCm38/mm10 assembly via UCSC under GCA_000001635.2. Source data are provided with this paper.

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

## Acknowledgements
We thank members of the Ramon Parsons Laboratory, Bert Vogelstein, Matthew Vander Heiden, and Lewis Cantley for discussions. We thank Saboor Hekmaty, Kevin Mohammed, and Ravi Sachidanandam from the Mount Sinai Oncological Sciences Sequencing Core Facility. We thank the Division of Colon and Rectal Surgery at Mount Sinai and Dr. Rachel Brody at the TCI Biorepository Core in Development for their assistance with sample collection. This study was supported by the following: National Institutes of Health grant F30CA243207 to R.W.Z., R35CA220491 to R.E.P., R01CA230854 to R.E.P., P30CA196521 to R.E.P, R01CA154683 to E.B., and R01CA218024 to E.B.

## Author contributions
R.W.Z. and R.E.P. conceptualized the study. R.W.Z., A.L.Z., S.O., J.X., T.C.M., A.B., J.H., A.P.T., B.Gr., B.Gi., D.D., Y.S., X.W., S.C., K.B., and J.A. performed the experimental studies and/or carried out the analysis. J.A., B.H., A.T., D.R.J., J.E.C., S.I., I.I.C.C., D.H., E.B., and R.E.P. supervised the study. R.S. and R.B. provided human samples. R.E.P. and R.W.Z. acquired funding.

## Competing interests
R.E.P. owns equity in Therapten. Ramon Parsons receives royalty payments from Cullgen and Therapten. R.E.P. gives industry-sponsored lectures at the Lurie Cancer Center and the University of Southern California Cancer Center. R.E.P. reports other activities with Columbia University and Regeneron Pharmaceuticals. S.H.I. consults for EXACT Sciences Corporation, and Geneoscopy. S.H.I. receives royalty payments from Bio-Rad Laboratories Inc. The remaining authors declare no competing interest.

## Additional information

[1]Department of Oncological Sciences, The Tisch Cancer Institute, Icahn School of Medicine at Mount Sinai, New York, NY 10029, USA. [2]Graduate School of Biomedical Sciences, Icahn School of Medicine at Mount Sinai, New York, NY 10029, USA. [3]Medical Scientist Training Program, Icahn School of Medicine at Mount Sinai, New York, NY 10029, USA. [4]Department of Genetics and Genomic Sciences, Icahn School of Medicine at Mount Sinai, New York, NY 10029, USA. [5]Department of Genetics and Genome Sciences, Case Western Reserve University, Cleveland, OH 44106, USA. [6]Division of Colon and Rectal Surgery, Department of Surgery, Icahn School of Medicine at Mount Sinai, New York, NY 10029, USA. [7]Metabolomics Core Resource Laboratory, NYU Langone Health, New York, NY 10016, USA. [8]Mass Spectrometry Core, Beth Israel Deaconess Medical Center, Boston, MA 02215, USA. [9]Mount Sinai Biorepository, Department of Pathology, Icahn School of Medicine at Mount Sinai, New York, NY 10029, USA. [10]Division of Gastroenterology, Department of Medicine, Icahn School of Medicine at Mount Sinai, New York, NY 10029, USA. [11]Institute for Cancer Genetics, Department of Genetics and Development, Columbia University Medical Center, New York, NY 10032, USA. [12]These authors contributed equally: Jia Xu, Tiphaine C. Martin. ✉e-mail: ramon.parsons@mssm.edu

