## [Peer Review File · Nature Communications]

A local tumor microenvironment acquired super-enhancer induces an oncogenic driver in colorectal carcinomaEditorial Note: This manuscript has been previously reviewed at another journal that is not operating a transparent peer review scheme. This document only contains reviewer comments and rebuttal letters for versions considered at *Nature Communications*

REVIEWERS' COMMENTS

Reviewer #1 (Remarks to the Author):

I recommend authors for carefully addressing points raised in my review. A couple of things that I am not convinced about:

1. I would recommend including Erouji et al data (IGVs) in the manuscript. This is an important independent validation given the batches (different datasets) of cell lines and tumor data in the first round of submission.
2. The effect is limited to HT29 cell line (not in DLD and other lines). And effect is most convincingly demonstrated in HT29 cell line versus xenograft which is derived from subcutaneous injection. Authors did not make an attempt to do an orthotopic injection in colonic epithelium. GEMM models are available: One of them is inducible APC, Kras and p53 mutant model from Depinho group (PMID: 28289141).
3. A convincing demonstration of how tumor microenvironment induces an (super)enhancer would be that this superenhancer is induced when a specific immune/TMEN cell type (for example, specific myeloid cells since NUDE mice lack T cells) is co-cultured with HT29, or alternatively, if HT-29 cells are treated with a specific cytokine.

Reviewer #2 (Remarks to the Author):

I think authors did a great job revising the manuscript and answering most of the critique. The paper is ready for Nat Communications as is.

Reviewer #3 (Remarks to the Author):

All key recommendations have been addressed.

A local tumor microenvironment acquired super-enhancer induces an oncogenic driver for efficient growth under oxidative conditions in colorectal carcinoma

Royce W. Zhou... Ramon E. Parsons

Reviewer #1 (Remarks to the Author):

I recommend authors for carefully addressing points raised in my review. A couple of things that i am not convinced about:

1. I would recommend including Erouji et al data (IGVs) in the manuscript. This is an important independent validation given the batches (different datasets) of cell lines and tumor data in the first round of submission.

Thank you for the comment. The data is now included in the manuscript as external, independent validation of findings from our own dataset.

The following has been added to Supplementary Figure S3 as panel A.

a PCA of H3K27ac signal at 2026 SEs in CRC ($n = 32$ independent tissue samples), CRC cell lines ($n = 11$ independent cell lines), and normal colon mucosa ($n = 14$ independent tissue samples). Data from Orouji et al, please see Methods for accessions.

The following has been added to Supplementary Figure S4 as panel B.

b H3K27ac ChIP-seq track at the *PDZK1IP1* SE in 15 representative independent primary CRCs and normal colon epithelium samples. Data from Orouji et al, please see Methods for accessions.

c H3K27ac ChIP-seq track at the *PDZK1IP1* SE in 10 representative independent primary CRCs and 8 representative CRC cell lines. Data from Orouji et al, please see Methods for accessions.

2. The effect is limited to HT29 cell line (not in DLD and other lines). And effect is most convincingly demonstrated in HT29 cell line versus xenograft which is derived from subcutaneous injection. Authors did not make an attempt to do an orthotopic injection in colonic epithelium. GEMM models are available: One of them is inducible APC, Kras and p53 mutant model from Depinho group (PMID: 28289141).

Thank you for the comment. We do not have expertise in using this particle orthotopic model. It may not work in our hands and although potentially interesting would not change the conclusions of the study. We have attempted a different orthotopic mouse model (PMID: 28459450), developed by our colleague Dr. Lukas Dow and Dr. Scott Lowe while at MSK. We spent a two month sabbatical in Dr. Dow's lab at Weill Cornell attempting to perform this model, however all mice failed to engraft any tumors even in their hands, a discrepancy that remains unexplained. This highlights the precarious success rates for implantation in orthotopic mouse models of CRC.

3. A convincing demonstration of how tumor microenvironment induces an (super)enhancer would be that this superenhancer is induced when a specific immune/TMEN cell type (for example, specific myeloid cells since NUDE mice lack T cells) is co-cultured with HT29, or alternatively, if HT-29 cells are treated with a specific cytokine.

Thank you for the comment—we agree this is a convincing demonstration. This data is included in the paper as is as Figure 5L, where HT29 cells in culture are treated with cytokines resulting in super-enhancer formation, reproduced below for easy viewing.

(L) H3K27ac ChIP-seq track of the *PDZK1IP1* SE (underlined) in unstimulated or TNF α , IFN γ , and IL-6 co-stimulated HT29 cells at 10 ng/mL for 16 hours. Y-axes of all ChIP-seq tracks are scaled to the same range [0-122].

Reviewer #2 (Remarks to the Author):

I think authors did a great job revising the manuscript and answering most of the critique. The paper is ready for Nat Communications as is.

Reviewer #3 (Remarks to the Author):

All key recommendations have been addressed.

We thank Reviewer #2 and Reviewer #3 again for your excellent contributions to our manuscript.